# Metabolic reprogramming by Acly inhibition using SB-204990 alters glucoregulation and modulates molecular mechanisms associated with aging

Alejandro Sola-García[1,11], María Ángeles Cáliz-Molina[1,11], Isabel Espadas[1,11], Michael Petr[2,3], Concepción Panadero-Morón[1], Daniel González-Morán[1], María Eugenia Martín-Vázquez[1], Álvaro Jesús Narbona-Pérez[1], Livia López-Noriega[1], Guillermo Martínez-Corrales[1], Raúl López-Fernández-Sobrino[1], Alejandro Castillo-Peña[1], Lina M. Carmona-Marin[2], Enrique Martínez-Force[4], Oscar Yanes[5,6], Maria Vinaixa[5,6], Daniel López-López[7,8,9], José Carlos Reyes[1], Joaquín Dopazo[7,8,9,10], Franz Martín[1,6], Benoit R. Gauthier[1,6], Morten Scheibye-Knudsen[2,3], Vivian Capilla-González[1] & Alejandro Martín-Montalvo[1,6✉]

ATP-citrate lyase is a central integrator of cellular metabolism in the interface of protein, carbohydrate, and lipid metabolism. The physiological consequences as well as the molecular mechanisms orchestrating the response to long-term pharmacologically induced Acly inhibition are unknown. We report here that the Acly inhibitor SB-204990 improves metabolic health and physical strength in wild-type mice when fed with a high-fat diet, while in mice fed with healthy diet results in metabolic imbalance and moderated insulin resistance. By applying a multiomic approach using untargeted metabolomics, transcriptomics, and proteomics, we determined that, in vivo, SB-204990 plays a role in the regulation of molecular mechanisms associated with aging, such as energy metabolism, mitochondrial function, mTOR signaling, and folate cycle, while global alterations on histone acetylation are absent. Our findings indicate a mechanism for regulating molecular pathways of aging that prevents the development of metabolic abnormalities associated with unhealthy dieting. This strategy might be explored for devising therapeutic approaches to prevent metabolic diseases.

[1] Andalusian Molecular Biology and Regenerative Medicine Centre-CABIMER, Universidad de Sevilla-CSIC-Universidad Pablo de Olavide, Seville 41092, Spain. [2] Center for Healthy Aging, Department of Cellular and Molecular Medicine, University of Copenhagen, Copenhagen, Denmark. [3] Tracked.bio, Copenhagen, Denmark. [4] Instituto de la Grasa (CSIC), Universidad Pablo de Olavide, Sevilla, Spain. [5] Universitat Rovira i Virgili, Department of electronic Engineering & IISPV, Tarragona, Spain. [6] CIBER de Diabetes y Enfermedades Metabólicas asociadas (CIBERDEM), Instituto de Salud Carlos III, Madrid, Spain. [7] Clinical Bioinformatics Area, Fundación Progreso y Salud (FPS), CDCA, Hospital Virgen del Rocio, c/Manuel Siurot s/n, 41013 Sevilla, Spain. [8] Computational Systems Medicine, Institute of Biomedicine of Seville (IBIS), Hospital Virgen del Rocio, Sevilla 41013, Spain. [9] Bioinformatics in Rare Diseases (BiER), Centro de Investigación Biomédica en Red de Enfermedades Raras (CIBERER), FPS, Hospital Virgen del Rocío, Sevilla 41013, Spain. [10] FPS/ELIXIR-es, Hospital Virgen del Rocío, Sevilla 42013, Spain. [11] These authors contributed equally: Alejandro Sola-García, María Ángeles Cáliz-Molina, Isabel Espadas. ✉email: alejandro.martinmontalvo@cabimer.es

During the last century, humans have reached the longest life expectancy in history. However, it is remarkable that the increase in life expectancy is associated with a plethora of age-related pathologies, and it is estimated that only ~45% of humans reaching 75 years of age describe a good quality of life[1]. These data indicate the need to develop novel effective therapies to prevent and treat age-related complications in order to promote healthy aging. From a biological point of view, aging is the consequence of the accumulation of a great variety of molecular and cellular modifications over time, which leads to a gradual decrease in physical, metabolic, and mental capacities, as well as an increased risk of disease and death. In mammals, most of the main age-related pathologies that shorten health and life expectancy, such as atherosclerosis, diabetes mellitus, sarcopenia, and hepatic steatosis, can be derived from improper metabolic control[2–4].

Acetyl-coenzyme A (Ac-CoA) is a central metabolite produced from all energy sources, such as amino acids, fatty acids, and carbohydrates[5]. This small molecule plays a key role in a great number of essential cellular processes, including the endogenous synthesis of fatty acids, cholesterol, and coenzyme Q[5,6]. Moreover, Ac-CoA is the universal acetyl group donor for protein acetylation, a post-translational modification that controls protein stability, localization, and function. ATP-citrate lyase (Acly) is the enzyme catalyzing the generation of nuclear-cytosolic Ac-CoA and oxaloacetate from citrate in the presence of ATP and coenzyme A. It is expressed ubiquitously, although greater expression is found in lipogenic tissues[7–9]. Moreover, several cancers have been shown to exhibit aberrantly increased Acly activity[10–12]. Acly is an essential enzyme, as demonstrated by the lack of viability of homozygous Acly knockout mice[8]. However, heterozygous Acly knockout mice are healthy and fertile and exhibit normal lipid metabolism. These results have supported that a partial loss of Acly activity is compatible with an optimal quality of life.

The central role of Acly in de novo lipogenesis has fostered a need to generate therapeutic strategies based on the use of pharmacological inhibitors as a hypolipidemic strategy for metabolic syndrome and cancer treatment[13]. Studies using short-term administrations of Acly inhibitors have reported promising results in refraining tumor growth or ameliorating metabolic parameters in mammals[14–17]. Remarkably, research using bempedoic acid, a dual Ampk activator/Acly inhibitor developed for the treatment of dyslipidemia and cardiometabolic disease[17], has provided positive results in lowering low-density lipoprotein cholesterol in clinical trials, and it is currently in the market[18–20]. However, the mechanisms that govern the handling of sustained Acly inhibition, and particularly how such mechanisms orchestrate long-term cellular reprogramming, remain to be elucidated.

Here we assessed the consequences of long-term exposure to the Acly inhibitor SB-204990 in mice[15]. We performed an unbiased multiomic approach integrating transcriptomics, proteomics, and untargeted metabolomics in murine hepatic tissue, given its central role in the maintenance of metabolic homeostasis. Analyses uncovered effects on energy metabolism, mitochondrial function, lipid metabolism, mTOR activity, as well as in the control of the folate cycle. Epigenetic studies indicated that these effects are not associated with global modulations in histone and non-histone protein acetylation. SB-204990 recapitulates certain effects of mTOR inhibitors in standard (STD)-fed mice and produces favorable effects in mice fed with a high-fat diet (HFD), which might provide therapeutic benefits against the current pandemic proportions of obesity-related metabolic disorders that predispose to unhealthy aging.

## Results

**Hepatic Acly expression is increased in aging mice.** To define age-dependent metabolic changes that contribute to promote aging processes as well as to cause premature death, we evaluated different parameters of glucose homeostasis at different ages in healthy STD-fed wild-type mice. Body weight and the weight of several tissues were greater in adult and old mice (Fig. 1a, b). The assessment of oral glucose tolerance indicated the absence of major age-dependent alterations in glucose or insulin levels during the tests among young and old mice. However, significant differences, specifically in adult vs. old mice, in circulating insulin levels were observed (Fig. 1c–f). The analysis of pyruvate tolerance and insulin sensitivity indicated a reduced ability to promote hepatic gluconeogenesis as well as severe insulin resistance in aged mice (Fig. 1g–j). In fasting conditions, old mice exhibited hyperinsulinemia while maintaining normoglycemia, producing a greater index of the homeostatic model assessment of insulin resistance (HOMA-IR) (Fig. 1k–m). These data indicated that the most prominent effects of aging involve a deterioration of functionality in insulin-target tissues in the control of glucose metabolism, suggesting that these effects could contribute to the aging phenotype. We then focused on determining whether the expression levels of Acly, a gene that occupies a central role in the carbohydrate-lipid metabolism interface, are altered in an age-dependent manner in metabolic tissues. Remarkably, old mice exhibited higher levels of Acly expression and enzyme activity in the liver (Fig. 1n and Supplementary Fig. S1a). These data suggest that hepatic Acly could play an important role in the development of metabolic alterations that occur during the aging process.

**SB-204990 improves metabolic and physical function in HFD-fed mice.** In physiological conditions, Acly has been proposed as the main producer of cytosolic Ac-CoA[5]. Cytosolic Ac-CoA is used for endogenous lipid production and for the production of malonyl-coenzyme A, an inhibitor of the carnitine palmitoyl-transferase I, required for fatty acid uptake into the mitochondria[21]. Radiolabeled [H3]-glucose incorporation into lipids was measured to indirectly assess liponeogenesis in primary hepatocytes isolated from male wild-type mice. Results indicated that SB-204990 elicits a dose-dependent inhibition of glucose-dependent de novo lipogenesis (Supplementary Fig. S2a). Cell death measured via ELISA and urea secretion indicated that toxicity occurs at concentrations greater than 10 μM, with marked toxicity at 100 μM (Supplementary Fig. S2b, c). Experiments in primary pancreatic islets isolated from wild-type mice indicated lower susceptibility to alterations promoted by SB-204990 in functional and viability tests when compared to hepatocytes (Supplementary Fig. S2d–f). Altogether, these results show that the in vitro inhibition of fatty acid biosynthesis mediated by SB-204990 inhibition of Acly is achievable at nontoxic concentrations (i.e., ~10 μM).

Pharmacokinetic studies were performed to determine circulating SB-204990 levels upon oral administration in mice. Results indicated that a single dose of 30 mg/kg of body weight of SB-204990 rendered a ~4 μM plasma concentration 2 h post-ingestion in the range of nontoxic concentrations used in in vitro studies (Fig. 2a). These data led us to estimate the optimal dose of SB-204990 for long-term in vivo studies at 0.25 mg compound/g of food (~4 g of food intake/day) in mice. A first cohort of young wild-type mice (5-week-old) was used to evaluate early metabolic responses and spontaneous activity (5-week treatment) by indirect calorimetry, indicating the lack of significant effects of SB-204990 (Supplementary Fig. S2g–p). A second cohort of wild-type mice was fed with a healthy STD or cholesterol-free diabetogenic HFD and were treated or not with

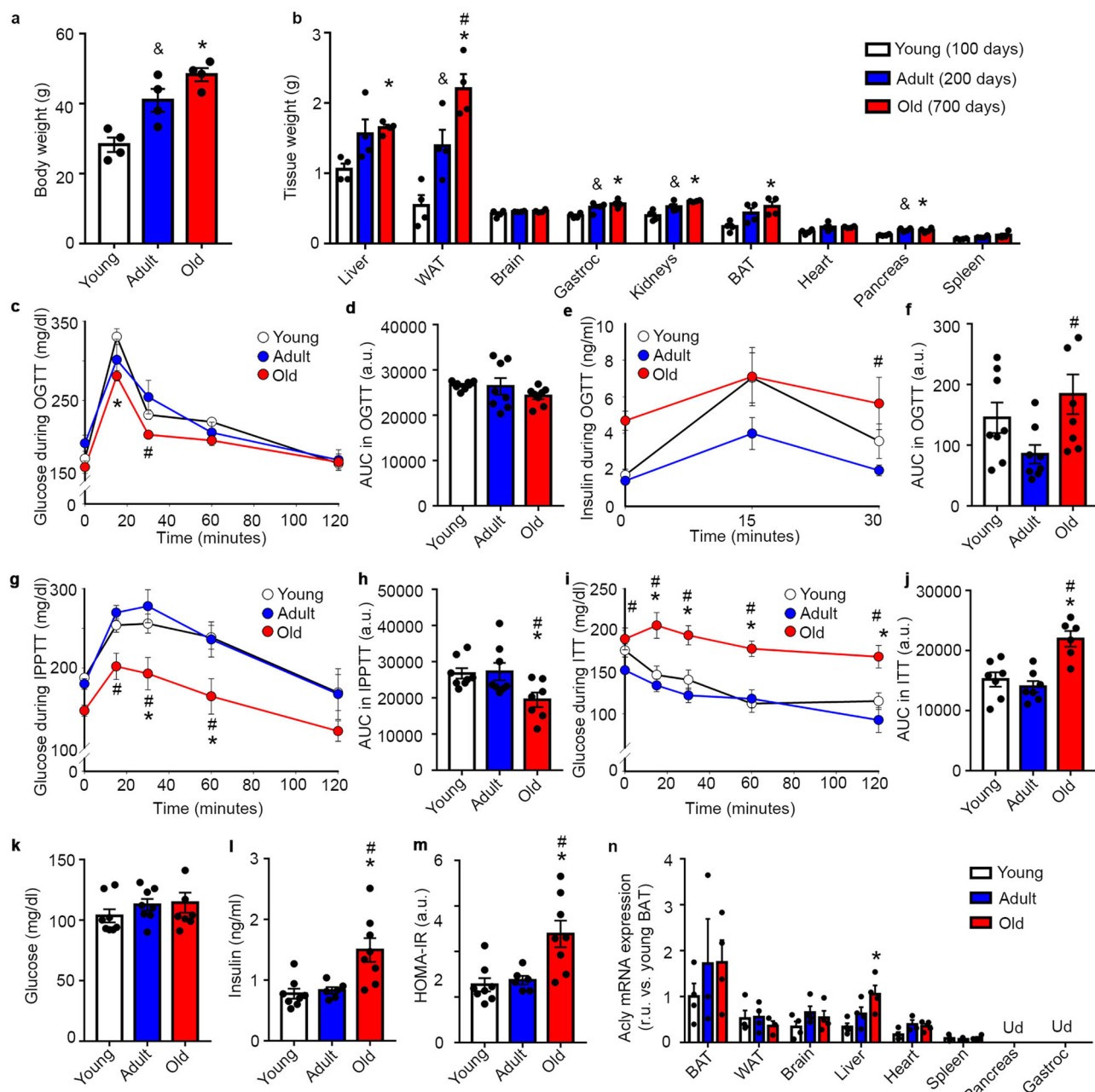

**Fig. 1 Hepatic Acly expression is increased in aging mice. a** Body weight. $n = 4$. One-way ANOVA. **b** Tissue weight. $n = 4$. One-way ANOVA. **c** OGTT. $n = 8$. Two-way ANOVA repeated measures. **d** Area under the curve (AUC) of the OGTT. $n = 8$. One-way ANOVA. **e** Insulin levels during an OGTT. $n = 8$. Two-way ANOVA repeated measures. **f** AUC of insulin levels during an OGTT. $n = 8$. One-way ANOVA. **g** IPPTT. $n = 8$ for young, $n = 8$ for adult, $n = 7$ for old. Two-way ANOVA repeated measures. **h** AUC of the IPPTT. $n = 8$ for young, $n = 8$ for adult, $n = 7$ for old. One-way ANOVA. **i** Insulin tolerance test (ITT). $n = 7$ for young, $n = 7$ for adult, $n = 6$ for old. Two-way ANOVA repeated measures. **j** AUC of the ITT. $n = 7$ for young, $n = 7$ for adult, $n = 6$ for old. One-way ANOVA. **k** Circulating glucose levels at 16 h of fasting. $n = 8$. One-way ANOVA. **l** Circulating insulin levels at 16 h of fasting. $n = 8$ for young, $n = 6$ for adult, $n = 8$ for old. One-way ANOVA. **m** HOMA-IR index. $n = 8$ for young, $n = 6$ for adult, $n = 8$ for old. One-way ANOVA. **n** Acly gene expression. $n = 3$–4. a.u.: arbitrary units. r.u.: relative units. BAT: Brown adipose tissue. Gastroc: Gastrocnemius. Ud: Under the threshold of detection. Data shown are the means ± SEM. *$p < 0.05$ Old vs. Young; #$p < 0.05$ Adult vs. Old; &$p < 0.05$ Young vs. Adult.

SB-204990 for 15 weeks starting at 26 weeks of age. As expected, in vivo treatment with the Acly inhibitor SB-204990 did not alter the enzymatic activity of Acly in liver tissue (Supplementary Fig. S2q)[15]. The body weight of STD-fed mice was lower than mice fed with HFD (Fig. 2b). HFD-SB-fed mice exhibited lower body weight starting at week 9 of treatment when compared to HFD-fed mice. Daily energy intake and lipid content in feces were not altered by SB-204990 supplementation (Supplementary Fig. S2r, s). However, fasting-induced energy intake was higher

in STD-SB when compared to their untreated counterparts (Supplementary Fig. S2t). We next investigated the effects of SB-204990 on glucose homeostasis. An oral glucose tolerance test (OGTT) and an intraperitoneal pyruvate tolerance test (IPPTT) indicated that, although animals fed with the same diets started with similar circulating glucose values, SB-204990 produced a mild impairment in glucose and pyruvate tolerance in STD-SB mice, while the administration of the compound to the HFD-fed mice had the opposite effect (i.e., improvement) when compared

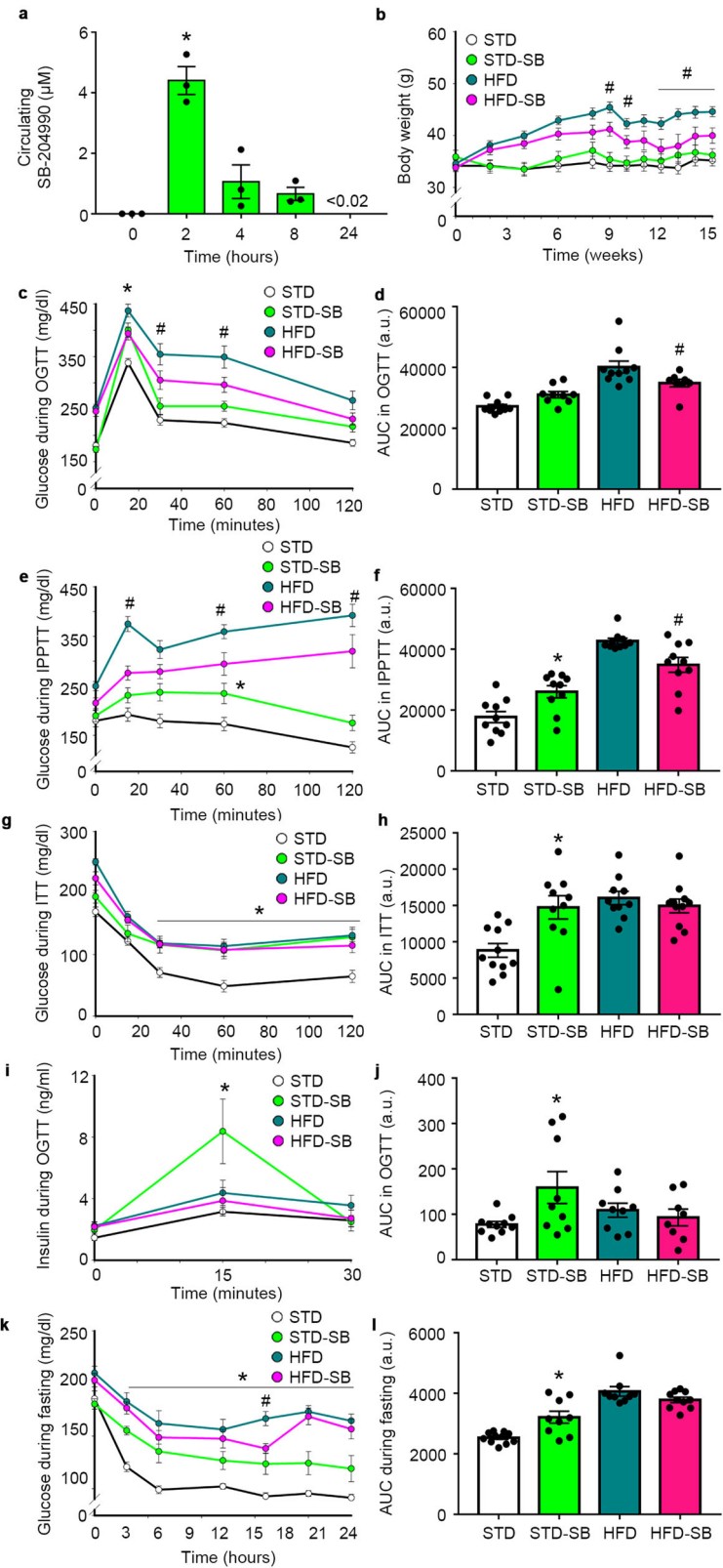

to untreated HFD, indicating a pivotal role of Acly in hepatic tissue (Fig. 2c–f). When challenged with an intraperitoneal insulin injection, all groups displayed a deficit in their responses when compared to the STD group (Fig. 2g, h). The determination of insulin levels during an OGTT indicated that STD-SB mice responded with a potent insulin secretion upon glucose challenge, suggesting that insulin resistance is the main underlying cause of

deregulated glucose homeostasis in STD-SB mice (Fig. 2i, j). HFD-SB mice exhibited similar insulin levels during the OGTT. The evaluation of glucose levels during a 24-hour fasting period indicated that STD-SB mice exhibited higher glucose levels when compared to STD mice, while HFD-fed mice exhibited similar levels irrespective of SB-204990 supplementation (Fig. 2k, l). Glucose levels and insulin levels in fasting conditions were greater

**Fig. 2 Long-term SB-204990 treatment improves metabolic health in mice fed with HFD and produces insulin resistance in STD-fed mice.**
**a** Pharmacokinetic evaluation of circulating SB-204990 levels in mice exposed to oral administration (30 mg/kg of body weight) of SB-204990. $n = 3$.
One-way ANOVA on Ranks. **b** Body weight over the course of the study. $n = 11$ for STD and STD-SB. $n = 10$ for HFD. $n = 11$ for HFD-SB weeks 0–13 and
$n = 9$ for weeks 13–15. Two-way ANOVA. **c** OGTT at week 13 of treatment. $n = 11$ for STD. $n = 9$ for STD-SB. $n = 10$ for HFD. $n = 8$ for HFD-SB. Two-way
ANOVA repeated measures. **d** AUC of the OGTT. $n = 11$ for STD. $n = 9$ for STD-SB. $n = 10$ for HFD. $n = 8$ for HFD-SB. Two-way ANOVA. **e** IPPTT at week
9 of treatment. $n = 10$. Two-way ANOVA repeated measures. **f** AUC of the IPPTT. $n = 10$. Two-way ANOVA. **g** ITT at week 11 of treatment. $n = 11$ for STD.
$n = 10$ for STD-SB. $n = 10$ for HFD. $n = 11$ for HFD-SB. Two-way ANOVA repeated measures. **h** AUC of the ITT. $n = 11$ for STD. $n = 10$ for STD-SB. $n = 10$ for
HFD. $n = 11$ for HFD-SB. Two-way ANOVA. **i** Insulin levels during an OGTT at week 10 of treatment. $n = 10$ for STD. $n = 9$ for STD-SB. $n = 9$ for HFD. $n = 8$
for HFD-SB. Two-way ANOVA repeated measures. **j** AUC of insulin levels during an OGTT. $n = 10$ for STD. $n = 9$ for STD-SB. $n = 9$ for HFD. $n = 8$ for HFD-
SB. Two-way ANOVA. **k** Glucose levels during 24 h fasting at week 14 of treatment. $n = 11$ for STD. $n = 9$ for STD-SB. $n = 9$ for HFD. $n = 9$ for HFD-SB.
Two-way ANOVA repeated measures. **l** AUC of glucose levels during 24 h fasting. $n = 11$ for STD. $n = 9$ for STD-SB. $n = 9$ for HFD. $n = 9$ for HFD-SB. Two-
way ANOVA. SB: SB-204990. STD: standard diet. HFD: high-fat diet. a.u.: arbitrary units. Data shown are the means ± SEM. $*p < 0.05$ different times vs.
time 0 or STD-SB vs. STD. $\#p < 0.05$ HFD-SB vs. HFD.

in STD-SB mice when compared to STD-fed mice, which produced a greater HOMA-IR index (Fig. 3a–c). In contrast, HFD-fed mice treated with SB-204990 exhibited lower fasting glucose, fasting insulin, and HOMA-IR values when compared to their HFD-fed counterparts (Fig. 3a–c). The percentage of glycated hemoglobin indicated a trend towards reduced levels in HFD-SB-fed mice when compared to HFD-fed mice ($p = 0.07$ in two-way ANOVA), with no differences in mice fed with a healthy diet (Fig. 3d).

In order to elucidate whether SB-204990-mediated Acly inhibition influences cholesterol homeostasis, circulating levels of low-density/very low-density lipoproteins (LDL/VLDL) and total cholesterol levels were measured (Fig. 3e, f). The administration of SB-204990 in HFD reduced circulating levels of LDL/VLDL and total cholesterol, while no changes were detected in the SB-204990-treated mice under STD when compared to their untreated counterparts. Remarkably, in both cases, values in the HFD-SB group were comparable to healthy-fed STD mice, demonstrating that SB-204990 prevents the deleterious effects of HFD on cholesterol homeostasis.

We next focused on determining the effects on motor function outcomes. Rotarod and wire hang physical performance were improved by SB-204990 in HFD, indicating the overall healthier status of these animals when compared to HFD-fed mice (Fig. 3g, h). Spatial memory was not altered by SB-204990 supplementation, irrespective of the diet (Supplementary Fig. S3a–f). Noticeably, the expression levels of cytokine Ccl2, but not those of other inflammatory or apoptotic markers (e.g., Caspase 3, Tnfα, Nfκb, and Gfap), were already upregulated specifically in the brains of untreated HFD-fed mice at the time of sacrifice (i.e., 15 weeks of treatment) (Supplementary Fig. S3g). At this time, STD-fed mice treated with SB-204990 exhibited increased white adipose tissue (WAT) mass, while mice fed with HFD supplemented with SB-204990 displayed reduced WAT mass when compared to their respective untreated diet (Fig. 3i). Histological examination of tissues indicated a massive reduction in lipid content in hepatic tissue and lower adipocyte size in HFD-SB-treated mice, while no major effects were observed in skeletal muscle and pancreatic sections (Fig. 3j, k). Similar to mice lacking WAT Acly, healthy-fed mice treated with SB-204990 exhibited greater mean adipocyte size in the WAT[22] (Fig. 3j, k). Serum levels of ALT (also known as Gpt) were increased in mice fed with HFD when compared to STD-fed mice, while serum AST (also known as Got) levels were not altered in HFD-fed mice. These makers of hepatic damage remained unaltered in mice treated with SB-2049990 in either STD or HFD (Fig. 3l, m). Altogether, these results show that SB-204990 has opposite effects depending on the diet provided. In healthy STD-fed mice, SB-204990 produces an impairment in glucose homeostasis due to insulin resistance, suggesting that a certain level of Acly activity is required to sustain metabolic health. However, under metabolically challenging conditions such as HFD feeding, the inhibition of Acly promoted by SB-204990 improves metabolic and physical health.

**Metabolic reprogramming induced by SB-204990 is independent of global changes in histone acetylation.** Intrigued by the contribution of Acly in cellular reprogramming in different metabolic conditions, we evaluated whether restricted Acly activity promoted by SB-204990 could alter histone acetylation, a global mechanism controlling transcription[23]. Incubation of AML12 hepatocyte cells with acetate, a substrate of the cytoplasmic Ac-CoA synthetase (also known as Acs, AceCS1, or Acss2) to produce cytosolic Ac-CoA, in the order of portal vein concentrations[24,25], produced maximal levels of histone acetylation on all core histones assessed (Fig. 4a and Supplementary Fig. S4a). Remarkably, an equimolar abundance of citrate, the substrate of the Acly, did not enhance histone acetylation, suggesting that citrate availability is not limiting acetylation events. Supraphysiologic levels of glucose, which could promote Ac-CoA generation via glycolysis-mitochondrial metabolism, did not result in consistent increases in histone acetylation, substantiating that net increases of histone acetylation might be more dependent on cytoplasmic Ac-CoA synthetase activity rather than Acly activity. Under these cellular conditions, SB-204990 treatment did not decrease histone acetylation levels (Fig. 4a and Supplementary Fig. S4a). Differences between the effects of acetate, citrate, and glucose and the lack of effects of SB-204990 support the notion that cytoplasmic Ac-CoA synthetase is a major source of Ac-CoA in the promotion of histone acetylation events that regulate chromatin structure in AML12 hepatocytes. In mouse liver, the analysis of histone acetylation, global levels of protein acetylation, and total Ac-CoA levels confirmed the lack of differences in SB-204990-treated mice fed with a healthy diet or with HFD (Fig. 4b–f). These data suggest that compensatory mechanisms potentiate Ac-CoA generation upon SB-204990-mediated Acly inhibition. Thus, modulation of global protein (including histone) acetylation is not the root cause of the metabolic effects of SB-204990.

**Multiomic analysis of diet-induced and SB-204990-induced hepatic reprogramming.** To gain insight into the underlying biology, we conducted a multiomic molecular profiling of the livers of mice treated with SB-204990 for 15 weeks. The transcriptional profile was evaluated in total RNA samples using cDNA microarray analyses. Untargeted GC/MS-based metabolomics was performed using the same groups of mice. Additionally, isobaric tag for absolute and relative quantification (iTRAQ) proteomics was conducted using samples from the same cohort of mice. We next developed a multiomic analysis to determine the major pathways that could explain the responses to

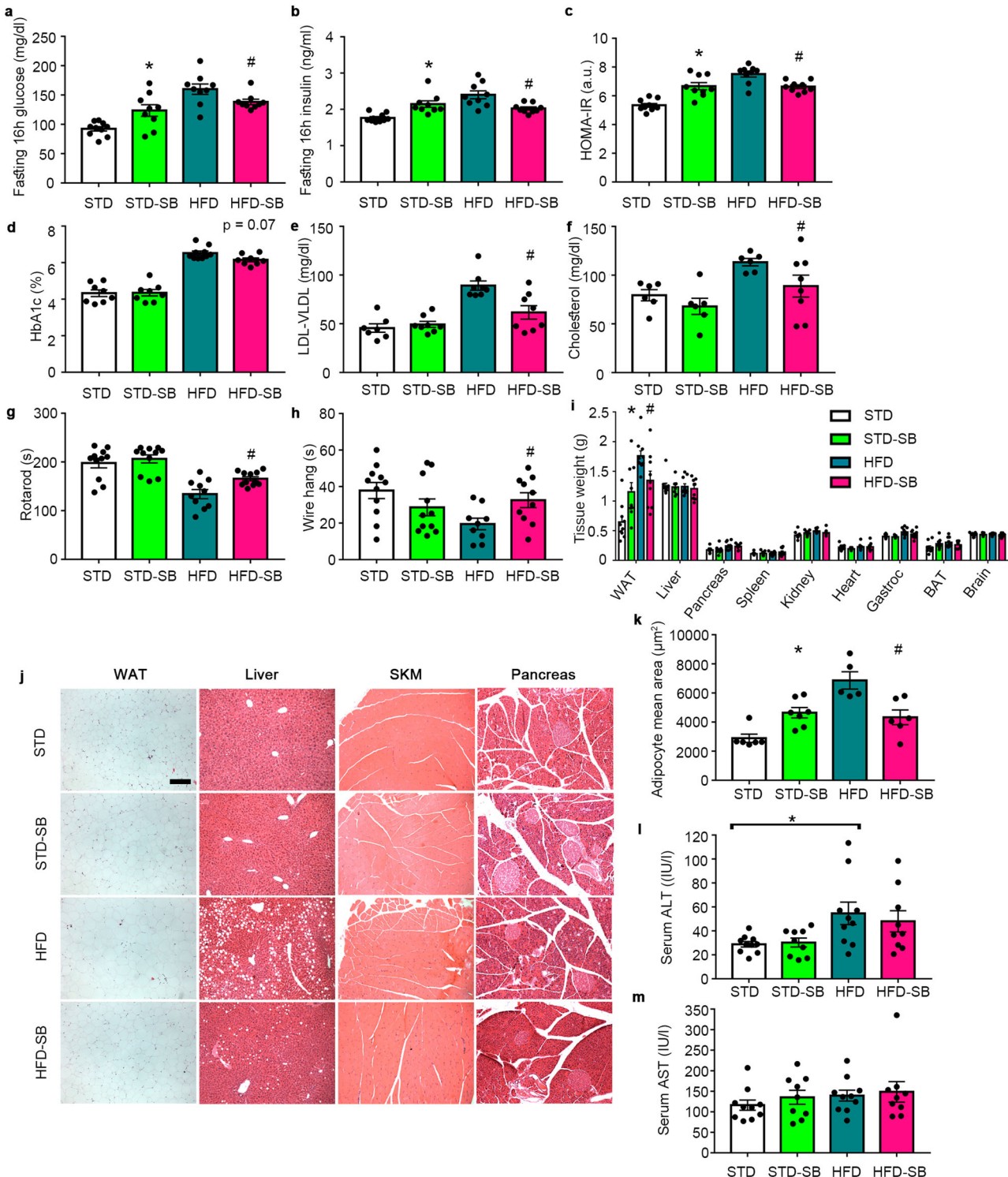

the different diets and the alterations promoted by SB-204990 treatment. As shown in the "butterfly" Venn diagrams, shared transcripts, metabolites, and proteins modulated by the different diets and SB-204990 treatment were identified (Fig. 5a). At the transcriptional level, we observed that a high proportion of significantly modulated transcripts ($p \le 0.05$) was specific to the different experimental conditions (425/891 HFD-SB vs. HFD; 502/948 HFD vs. STD; 539/1024 HFD-SB vs. STD; 557/814 STD-SB vs. STD). The analysis of GC/MS-based metabolomic changes revealed similar results (1/3 HFD-SB vs. HFD; 1/3 HFD vs. STD; 3/3 HFD-SB vs. STD; 7/10 STD-SB vs. STD). The analysis of

iTRAQ ($p \le 0.05$) proteomics revealed a lower degree of specificity (141/999 HFD-SB vs. HFD; 13/751 HFD vs. STD; 50/277 HFD-SB vs. STD; 5/144 STD-SB vs. STD). Remarkably, the vast majority of modulations shared between the HFD-SB vs. HFD and HFD vs. STD were found to be reciprocal (205/206 transcripts; 1/1 metabolite; 731/731 proteins), indicating that changes promoted by HFD are partially reverted by SB-204990 in mice fed with HFD. We next performed a multiomic analysis to determine modulations promoted by SB-204990 treatments independent of the diet (HFD or STD) using the subsets (drop shape in butterfly Venn diagrams) of transcripts, metabolites, and proteins (Fig. 5b).

**Fig. 3 SB-204990 improves physical health in mice fed with HFD. a** Circulating glucose levels in fasting conditions at week 14 of treatment. $n = 10$ for STD. $n = 9$ for STD-SB. $n = 9$ for HFD. $n = 9$ for HFD-SB. Two-way ANOVA. **b** Circulating insulin levels in fasting conditions at week 14 of treatment. $n = 10$ for STD. $n = 9$ for STD-SB. $n = 9$ for HFD. $n = 9$ for HFD-SB. Two-way ANOVA. **c** HOMA-IR index at week 14 of treatment. $n = 10$ for STD. $n = 9$ for STD-SB. $n = 9$ for HFD. $n = 9$ for HFD-SB. Two-way ANOVA. **d** Percentage of glycated hemoglobin (HbA1c) in blood at week 15 of treatment. $n = 8$ for STD. $n = 8$ for STD-SB. $n = 10$ for HFD. $n = 8$ for HFD-SB. Two-way ANOVA. **e** LDL/VLDL levels in serum at week 15 of treatment. $n = 7$ for STD. $n = 8$ for STD-SB. $n = 8$ for HFD. $n = 8$ for HFD-SB. Two-way ANOVA. **f** Total cholesterol levels in serum at week 15 of treatment. $n = 6$ for STD. $n = 6$ for STD-SB. $n = 6$ for HFD. $n = 8$ for HFD-SB. Two-way ANOVA. **g** Time to fall in an accelerating rotarod at week 8 of treatment. $n = 11$ for STD. $n = 11$ for STD-SB. $n = 10$ for HFD. $n = 11$ for HFD-SB. Two-way ANOVA. **h** Time to fall in wire hang test at week 10 of treatment. $n = 11$ for STD. $n = 11$ for STD-SB. $n = 9$ for HFD. $n = 10$ for HFD-SB. Two-way ANOVA. **i** Tissue weights. $n = 10$ for STD. $n = 9$ for STD-SB. $n = 10$ for HFD. $n = 9$ for HFD-SB. Two-way ANOVA. **j** Representative images of hematoxylin and eosin staining on several tissues. Scale 200 μm. $n = 6$ for STD. $n = 7$ for STD-SB. $n = 5$ for HFD. $n = 6$ for HFD-SB. **k** Adipocyte mean area. $n = 6$ for STD. $n = 7$ for STD-SB. $n = 5$ for HFD. $n = 6$ for HFD-SB. Two-way ANOVA. **l** Serum ALT levels in fasting conditions at week 14 of treatment. $n = 10$ for STD. $n = 9$ for STD-SB. $n = 10$ for HFD. $n = 9$ for HFD-SB. Two-way ANOVA. **m** Serum AST levels in fasting conditions at week 14 of treatment. $n = 10$ for STD. $n = 9$ for STD-SB. $n = 10$ for HFD. $n = 9$ for HFD-SB. Two-way ANOVA. SB: SB-204990. STD: standard diet. HFD: high-fat diet. BAT: Brown adipose tissue. Gastroc: gastrocnemius. SKM: skeletal muscle. a.u.: arbitrary units. Data shown are the means ± SEM. Unless otherwise highlighted, *$p < 0.05$ STD-SB vs. STD. #$p < 0.05$ HFD-SB vs. HFD.

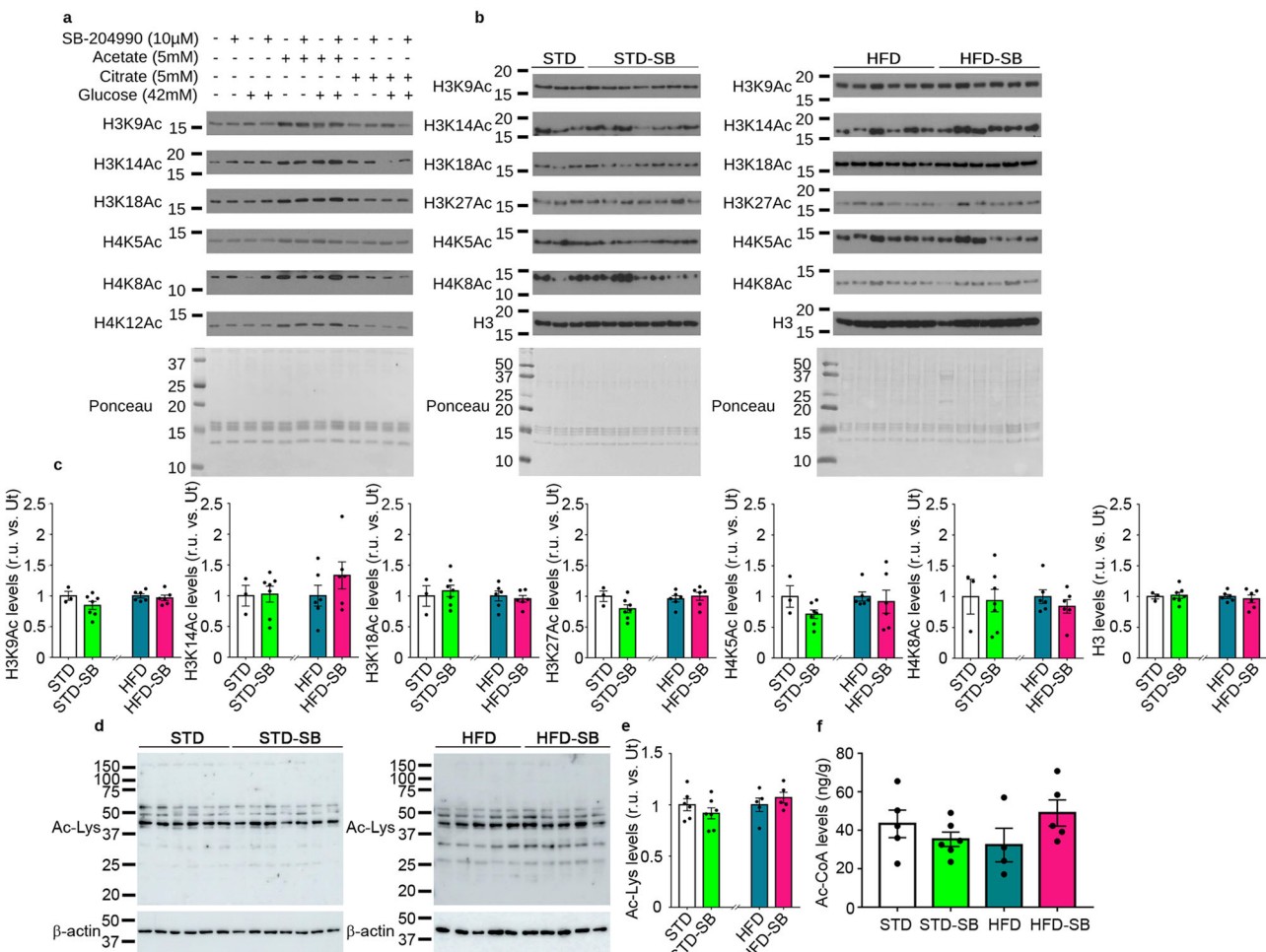

**Fig. 4 Hepatic levels of histone acetylation are not altered by SB-204990. a** AML12 cells were cultured under standard conditions and were treated with indicated doses of acetate, citrate, and SB-204990 for 16 h. Glucose concentration in basal conditions is 17 mM, and supplemented is 42 mM. Acid histone isolations were performed, and histone acetylation levels were analyzed by western blot. Representative images of western blots and Ponceau staining are shown. $n = 3$. **b** Acid histone isolations were performed, and histone acetylation levels were analyzed by western blot in the livers of mice treated or not with SB-204990 for 15 weeks. Representative images of western blots and Ponceau staining are shown. $n = 3$ for STD. $n = 7$ for STD-SB. $n = 6$ for HFD. $n = 6$ for HFD-SB. **c** Quantification of histone acetylation levels of several residues is depicted. $n = 3$ for STD. $n = 7$ for STD-SB. $n = 6$ for HFD. $n = 6$ for HFD-SB. Student's $t$-test. **d** Representative images of western blots of protein lysine acetylation are shown. Molecular weight markers are depicted on the left. $n = 6$ for STD. $n = 7$ for STD-SB. $n = 5$ for HFD. $n = 5$ for HFD-SB. **e** Densitometric quantification of hepatic protein lysine acetylation by western blot. $n = 6$ for STD. $n = 7$ for STD-SB. $n = 5$ for HFD. $n = 5$ for HFD-SB. Student's $t$-test. **f** Total Ac-CoA levels in hepatic tissue. $n = 5$ for STD. $n = 6$ for STD-SB. $n = 4$ for HFD. $n = 5$ for HFD-SB. Two-way ANOVA. SB: SB-204990. STD: standard diet. HFD: high-fat diet. Ac-Lys: Acetylated lysine. r.u.: relative units. Ut: untreated. Data shown are the means ± SEM.*$p < 0.05$ STD-SB vs. STD. #$p < 0.05$ HFD-SB vs. HFD.

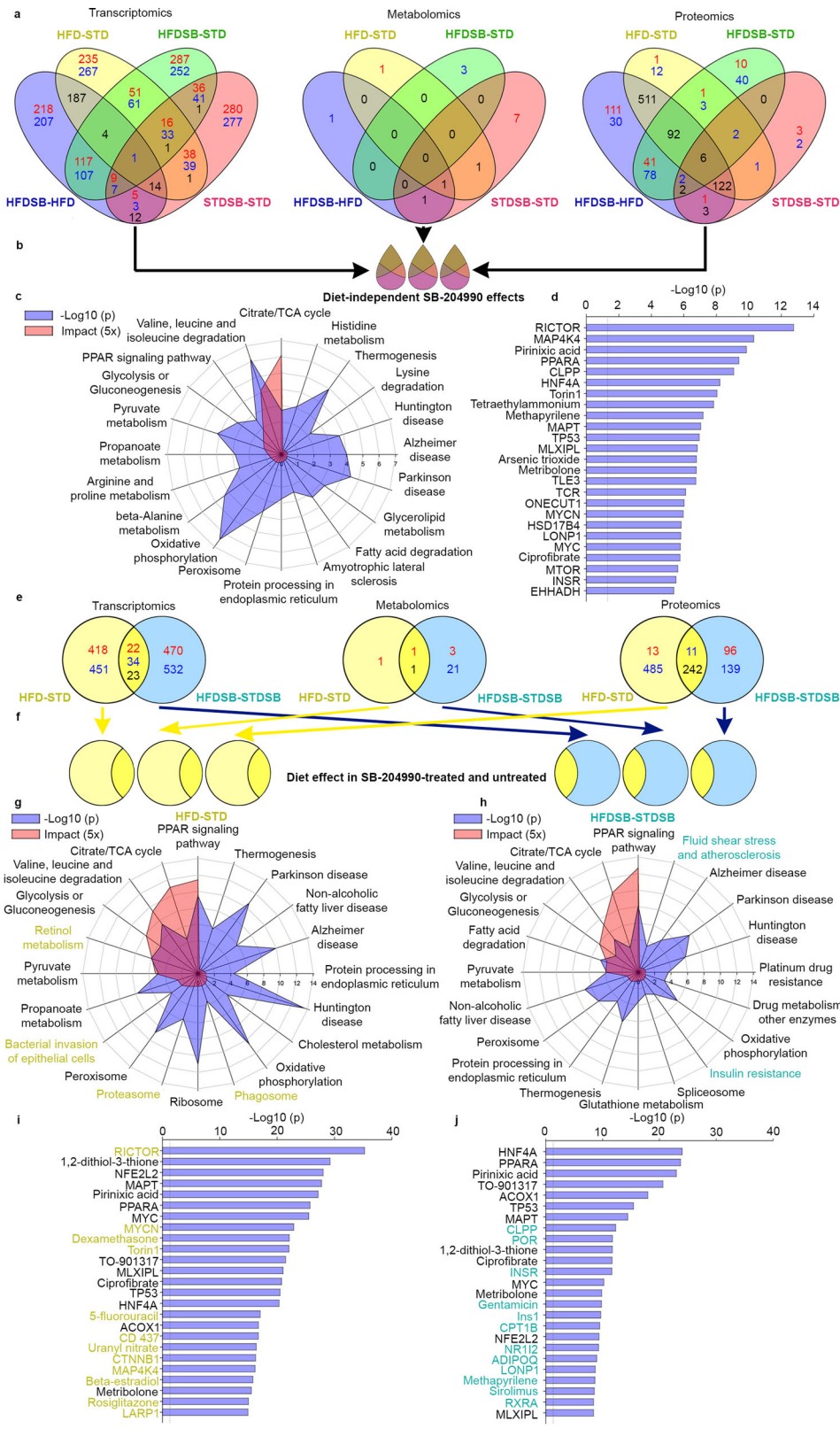

Interestingly, among the common upregulated transcripts Srebf2, a master regulator of cholesterol biosynthesis, was included (Supplementary Data 1, Supplementary Table S1, and Supplementary Data 2). Using the joint pathway analysis (JPA) module from Metaboanalyst 5.0[26], we found that the TCA cycle, oxidative phosphorylation, branched amino acid degradation, and Ppar signaling were among the most significant ranked with the

highest impact modulated pathways (Fig. 5c). Effects on mitochondrial function and fatty acid metabolism were also highlighted using the ToxList module of the Ingenuity Pathway Analysis (IPA) platform (Supplementary Fig. S5a). The analysis of Upstream regulators using IPA revealed Rictor and PPARα among the most significant effectors of SB-204990-induced modulations irrespective of the diet (Fig. 5d). We next took a

**Fig. 5 Multiomic analysis of the effects of SB-204990 in the liver; a major role in longevity pathways. a** Multiomics analysis in liver samples using transcriptomics, proteomics, and GC/MS-based metabolomics. Significantly altered transcripts, proteins, and metabolites are shown in Venn diagrams. Upregulation (red), downregulation (blue), and reciprocal regulation (black). **b** Drop shape analysis indicating diet-independent significant modulation in transcripts, proteins, and metabolites altered by SB-204990 in STD and HFD. **c** The JPA from MetaboAnalyst 5.0 was used to generate plots indicating the *p*-value and impact of the top 20 pathways with the lowest *p*-value according to the analytical scheme shown in Fig. 7b. **d** The Upstream Regulators module of IPA was used to generate plots depicting the top 25 upstream regulators with the lowest *p*-value according to the analytical scheme shown in Fig. 7b. **e** Significantly altered transcripts, proteins, and metabolites are shown in Venn diagrams depicting the effects the HFD vs. STD-responsive transcriptomic, metabolomic, and proteomic responses in the liver of mice treated or not with SB-204990. Upregulation (red), downregulation (blue), and reciprocal regulation (black). **f** Diagram indicating significant transcriptomic, metabolomic, and proteomic responses promoted by HFD in the liver of mice treated or not with SB-204990. **g** JPA plots indicating the *p*-value and impact of the top 20 pathways with the lowest *p*-value of HFD vs. STD according to the analytical scheme shown in Fig. 7f. Specific pathways are highlighted in yellow. **h** JPA plots indicating the *p*-value and impact of the top 20 pathways with the lowest *p*-value of HFD-SB vs. STD-SB according to the analytical scheme shown in Fig. 7f. Specific pathways are highlighted in blue. **i** Upstream Regulators plots depicting the top 25 upstream regulators with the lowest *p*-value according to the analytical scheme shown in Fig. 7f. Specific regulators are highlighted in yellow. **j** Upstream Regulators plots depicting the top 25 upstream regulators with the lowest *p*-value according to the analytical scheme shown in Fig. 7f. Specific regulators are highlighted in blue. Transcriptomic analysis; $n = 4$ for STD. $n = 4$ for STD-SB. $n = 3$ for HFD. $n = 3$ for HFD-SB. iTRAQ proteomic analysis $n = 4$. Metabolomic GC/MS analysis; $n = 5$ for STD. $n = 7$ for STD-SB. $n = 6$ for HFD. $n = 6$ for HFD-SB. SB: SB-204990. STD: standard diet. HFD: high-fat diet.

different approach to understand the biology behind the responses to STD/HFD feeding and SB-204990 treatment in liver tissue. We evaluated the HFD vs. STD-responsive transcriptomic, metabolomic, and proteomic responses in the liver of mice treated or not with SB-204990. Then, we looked for signatures that were specific and common to untreated or SB-204990-treated mice[27]. Remarkably, only 56 transcripts, 1 metabolite, and 11 proteins were significantly altered in the same direction, indicating that SB-204990 elicits differential responses in STD and HFD diets (Fig. 5e, f). JPA analysis revealed that HFD specifically harnessed responses related to inflammatory pathways (bacterial invasion and phagosome), Ampk signaling, and retinol metabolism, whereas mice treated with SB-204990 reflected specific effects in atherosclerosis, insulin resistance, and metabolism of certain amino acids (Phenylalanine, Arginine, and Proline) (Fig. 5g, h and Supplementary Data 3). The analysis of upstream regulators using IPA highlighted Rictor as the most significantly altered regulator in untreated mice (Fig. 5i). Interestingly, Rictor was not found among the top regulators in mice treated with SB-204990 (Fig. 5j). These results indicate that the Acly inhibitor SB-204990 modulates central pathways of cellular metabolism.

**Cholesterogenesis and one-carbon metabolism/folate cycle are major effectors of SB-204990 action.** In order to investigate the impact of the Acly inhibitor SB-204990 in each diet, we next investigated modulation in healthy-fed (STD diet) mice as well as in unhealthy-fed (HFD) mice treated or not with SB-204990 (Fig. 6a, b and Supplementary Fig. S6a–f). Multiomic analyses using transcriptomics, metabolomics (LC/MS- and GC/MS-based), and proteomics indicated that pyruvate metabolism, cholesterol metabolism, and oxidative phosphorylation were among the most significantly altered pathways in STD mice treated with SB-204990 (Fig. 6a and Supplementary Table S2). A similar analysis in mice fed HFD exposed to SB-204990 indicated that mitochondria-related alterations/diseases exhibited the highest significance (Fig. 6b and Supplementary Data 4). Citrate/ TCA cycle and one-carbon pool by folate, which have been identified as key metabolic nodes to enhance healthspan and lifespan, were highlighted with the highest impact by Metaboanalysist JPA in both diets (Fig. 6a, b)[27,28]. ToxList using the IPA platform, as well as wikipathways analysis using the Transcriptional Analysis Console (TAC) platform, supported alterations in fatty acid metabolism, mitochondrial function, one-carbon metabolism, and methylation (Supplementary Fig. S6a–d). Upstream IPA analysis indicated that Insulin receptor (Insr) and Ppar-α were among the most significant regulators altered in

STD-SB mice (Supplementary Fig. S6e), while Rictor was found in HFD-SB mice (Supplementary Fig. S6f). Alterations in pathways controlling the synthesis and utilization of different kinds of fats drove us to determine hepatic changes in lipid species promoted by SB-204990. The analysis of hepatic cholesterol esters indicated a robust reduction in both SB-treated STD and SB-treated HFD-fed mice (Fig. 6c). Taking advantage of our unbiased multiomic approach, we observed that, similar to effects of statins, reduced hepatic cholesterol was associated with increased levels of cholesterogenic machinery, indicating effective inhibition of downstream pathways of Acly activity (Fig. 6d–f, Supplementary Data 1)[29,30]. The global analysis of hepatic fatty acids indicated minor differences in fatty acid species, revealing a slight reduction in palmitic acid in STD-SB mice and a modest increase in vaccenic acid in HFD-SB mice (Fig. 6g). Consequently, the analysis of triglyceride species revealed a specific decrease in triglycerides containing two palmitic acids (POP and PLP) in STD-SB mice (Fig. 6h). In HFD-SB mice we found greater hepatic levels of several triglyceride species containing two saturated fatty acids (POP, PLS) and lower levels of triglycerides containing polyunsaturated fatty acids (POL, PLL, and OOL), an effect that has been observed in Acly depleted LNCaP cells (Fig. 6h)[31]. Analysis of polar lipid species indicated reductions of global phosphatidylserine levels in STD-SB mice (Supplementary Fig. S6g). The analysis of modulated hydrophobic species in our untargeted LC/MS metabolomics indicated increased levels of several long-chain phosphatidylcholines species, while short-chain phosphatidylcholines were decreased (Fig. 6i). Moreover, alterations in diglyceride and phosphatidylinositol species were also detected, indicating a profound modulation in the lipidome (Fig. 6i). Remarkably, an overall agreement with these trends was also found in SB-204990 treated mice fed with HFD (Fig. 6i). In addition, we observed a robust reduction of pteridine and betaine precursors related to folate cycle and one-carbon metabolism and a concomitant increased in transmethylated carnitine in STD-fed mice treated with SB-204990 (Fig. 6j). Of note, glycerol 3-phosphate, a precursor in the synthesis of glycerolipids, was increased in the liver of STD-SB mice (Fig. 6j). Interestingly, SB-204990 failed to affect these metabolite levels in mice fed with HFD. These data suggest that SB-204990-induced modulations in the folate cycle are more prominent in healthy-fed conditions, revealing differences in how this pathway is engaged under different feeding regimens. Overall, results shown revealed that SB-204990 produces modulations in pathways related to improved survival and healthspan, which include the metabolism of several amino acids and lipids[27,28].

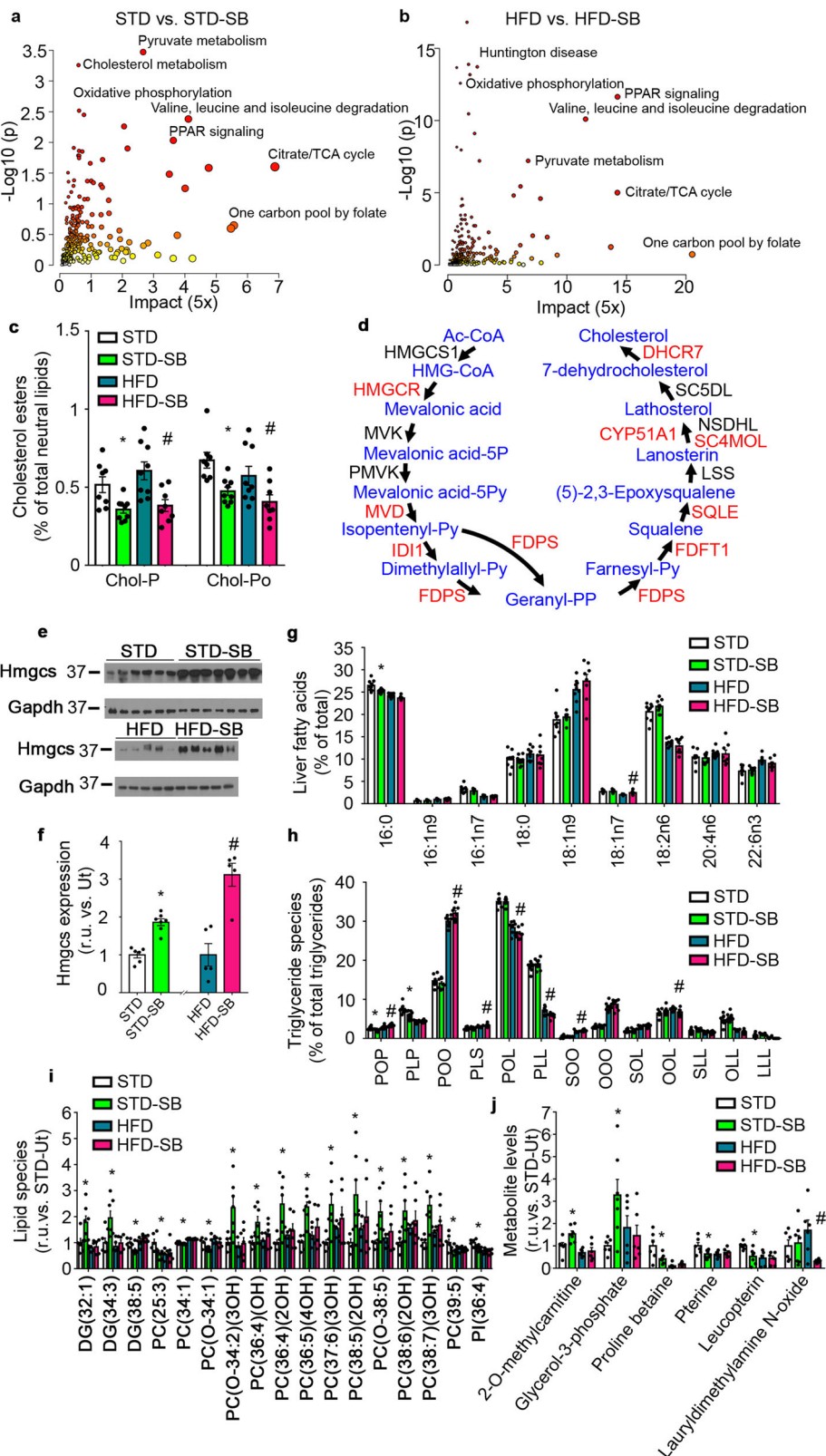

**SB-204990 targets hepatic mitochondrial function**. Multiomic analysis revealed a marked effect of SB-204990 in pathways related to cellular energetics and mitochondrial pathways in STD and HFD feeding regimens (e.g., TCA cycle, pyruvate metabolism, and oxidative phosphorylation) (Fig. 6a, b, Supplementary Table S2, Supplementary Data 4, and Supplementary Fig. S7a, b). Contrary to the putative role of Acly in the production of Ac-CoA

for de novo fatty acid synthesis and despite the fact that glucose incorporation into lipids was reduced by SB-204990, we found that primary hepatocytes exhibited a rapid increase in lipid content upon treatment with SB-204990 (Fig. 7a, b and Supplementary Fig. S2a). Interestingly, similar effects (e.g., larger lipid droplets/adipocyte size) have been observed in healthy-fed mice and cell cultures depleted of Acly activity (Fig. 3j, k)[22,31–33].

**Fig. 6 SB-204990 modulates fatty acid, cholesterol, and phospholipid metabolism. a, b** Multiomics analysis in liver samples using transcriptomics, proteomics, and GC/MS- and LC/MS-based metabolomics. The JPA from MetaboAnalyst 5.0 was used to generate plots indicating the *p*-value and impact of significantly modulated pathways. **a** STD-SB vs. STD. Transcriptomic analysis; *n* = 4. iTRAQ proteomic analysis *n* = 4. Metabolomic GC/MS analysis; *n* = 5 for STD. *n* = 7 for STD-SB. Metabolomic LC/MS analysis; *n* = 6 for STD. *n* = 7 for STD-SB. **b** HFD-SB vs. HFD. Transcriptomic analysis; *n* = 3. iTRAQ proteomic analysis *n* = 4. Metabolomic GC/MS *n* = 6. Metabolomic LC/MS analysis *n* = 6. **c** Hepatic cholesterol esters. Chol-P: Cholesteryl palmitate. Chol-Po: Cholesteryl palmitoleate *n* = 8 for STD. *n* = 9 for STD-SB. *n* = 9 for HFD. *n* = 8 for HFD-SB. Two-way ANOVA. **d** Integration of multiomic analysis in cholesterol biosynthetic pathway in HFD-SB mice. Red indicates upregulation. Black indicates no significant differences. **e** Immunoblots for Hmgcs and Gapdh from liver homogenates. **f** Densitometric analyses of immunoblots shown in panel E. *n* = 6 for STD. *n* = 7 for STD-SB. *n* = 5 for HFD. *n* = 5 for HFD-SB. Student's *t*-test. **g** Hepatic fatty acids. *n* = 9 for STD. *n* = 9 for STD-SB. *n* = 9 for HFD. *n* = 8 for HFD-SB. Two-way ANOVA. **h** Hepatic triglyceride species. P: Palmitic acid. O: Oleic acid. L: Linoleic acid. S: Stearic acid. *n* = 8 for STD. *n* = 9 for STD-SB. *n* = 9 for HFD. *n* = 8 for HFD-SB. Two-way ANOVA. **i** Analysis of significantly altered hydrophobic metabolites by LC/MS. *n* = 6 for STD. *n* = 7 for STD-SB. *n* = 6 for HFD. *n* = 6 for HFD-SB. Two-way ANOVA. **j** Analysis of significantly altered hydrophilic metabolites by LC/MS. *n* = 6 for STD. *n* = 7 for STD-SB. *n* = 6 for HFD. *n* = 6 for HFD-SB. Two-way ANOVA. SB: SB-204990. STD: standard diet. HFD: high-fat diet. r.u.: relative units. Ut: untreated. Data shown are the means ± SEM. *$p < 0.05$ STD-SB vs. STD. #$p < 0.05$ HFD-SB vs. HFD.

Incubation of primary hepatocytes with acetate or citrate did not increase lipid content (Fig. 7c and Supplementary Fig. S7c). Culture conditions mimicking hyperglycemia showed a trend towards increased lipid accumulation that was statistically significant only when supplemented with acetate. Under basal glucose conditions, SB-204990 produced net increases in lipid content even when culture media was supplemented with citrate and acetate. Lipid accumulation under conditions of SB-204990-mediated Acly inhibition might be explained by restricted metabolic activity. MTT (3-(4,5-dimethylthiazol-2-yl)-2,5-diphenyltetrazolium bromide) test indicated that, under basal conditions, SB-204990 reduces cellular metabolism (Fig. 7d). Remarkably, acetate supplementation did not restore the metabolic activity of primary hepatocytes treated with SB-204990, supporting previous observations indicating that restricted Acly activity cannot be fully compensated with acetate supplementation[22]. Similar to treatments using SB-204990, experiments using bempedoic acid, a Food and Drug Administration (FDA)-approved Acly inhibitor, showed increased hepatocyte lipid content and reduced MTT activity, supporting that these effects rely on Acly inhibition (Fig. 7e, f). Intrigued by the potential effect of SB-204990 on mitochondrial performance, which could potentiate lipid accumulation, we evaluated mitochondrial functional dynamics in primary hepatocytes treated with SB-204990. SB-204990 reduced basal and maximal oxygen consumption rates (OCR) under standard culture conditions (e.g., low glucose; LG), while specifically maximal OCR was diminished in cells cultured on high glucose (25 mM) treated with SB-204990 (Fig. 7g). In standard and high glucose media, SB-204990 promoted reduction of extracellular acidification rate (ECAR) in cells challenged with oligomycin and FCCP, suggesting reduced glycolysis (Fig. 7h). Similarly, experiments using bempedoic acid also showed a reduction on maximal OCR in hepatocytes cultured with standard (5 mM) and high glucose (25 mM) concentration and restricted oligomycin-induced and FCCP-induced ECAR (Supplementary Fig. S7d, e). Direct evaluation of glycolysis in primary hepatocytes indicated that SB-204990 produced net decreases in ECAR in both basal and high glucose conditions (Fig. 7i). These results support the allosteric inhibition of glycolysis by intracellular accumulation of citrate[34]. We next determined whether SB-204990 could elicit effects in the mitochondria in vivo. Freshly isolated liver mitochondria of mice treated with SB-204990 (30 mg/kg of body weight; oral gavage; 3 h of treatment) exhibited a blockade and restricted increase of ADP-induced and FCCP-induced OCR, respectively (Fig. 7j). These data indicate a rapid effect of SB-204990 in mitochondrial uncoupling and suggest that factors involved in energy storage and utilization could be modulated under conditions of Acly inhibition mediated by SB-204990.

We next focused on determining modulations promoted by SB-204990 in the liver of mice fed with a healthy STD or an obesogenic HFD. We found that pThr 172 Ampk phosphorylation was reduced in the liver of mice treated for 15 weeks with SB-204990, irrespective of the diet, indicating a rich energetic status. Moreover, we found markers of restricted mTOR signaling and increased Sirt1 expression in STD-SB-204990 mice, whereas unaltered mTOR signaling and increased Sirt1 and Nmnat expression were found in HFD-SB-204990 mice (Fig. 7k and Supplementary Fig. S7f). These results suggest that enhanced Sirt1 expression or activity via greater $NAD^+$ availability and restricted mTOR activity (specifically in STD-SB mice) could be concomitant to reduced Ampk activity[27,35]. Importantly, markers of mitochondrial content and biogenesis were reduced in the liver of mice treated with SB-204990 fed with a healthy diet, while expression levels of Acaa2, involved in mitochondrial β-oxidation, were increased in both STD-fed and HFD-fed mice treated with SB-204990 (Fig. 7k and Supplementary Fig. S7f). Modulations of mitochondrial functionality could alter stress resistance and oxidative damage accumulation[36]. Therefore, we next sought to determine expression levels of markers of the DNA damage response and lipid peroxidation. Interestingly, liver lysates of STD-SB mice exhibited reduced levels of Gadd153, while the livers of HFD-SB exhibited reduced lipid peroxidation (lysine 4-hydroxynonenal). Remarkably, the expression of the antioxidant protein Sod1 was increased in the livers of STD-SB and HFD-SB mice, while markers of autophagy exhibited divergent effects (Supplementary Fig. S7g, h). Under healthy feeding conditions, Fas levels were reduced, supporting the notion that reduced mitochondrial metabolism might have a greater specific weight promoting lipid accumulation in vivo than alterations on de novo lipogenesis (Fig. 7k and Supplementary Fig. S7f). In order to determine effects occurring early upon exposure to the Acly inhibitor SB-204990, we determined modulations on pathways altered in the liver of mice using cell cultures of primary hepatocytes. Similar to hepatic tissue of healthy-fed mice, we found that pThr 172 Ampk phosphorylation and pSer 79 Acc phosphorylation were reduced in primary hepatocytes treated with SB-204990 for 16 h, whereas alterations on markers of Sirt1 or mTOR activity were absent (Fig. 7l and Supplementary Fig. S7i). Primary hepatocytes were treated simultaneously with SB-204990 and the Ampk activator metformin (Fig. 7m–o and Supplementary Fig. S7j–l). Metformin normalized metabolic activity and lipid levels in SB-204990-treated hepatocytes, indicating that these effects require the blockade of Ampk activity. Moreover, metformin further exaggerated mitochondrial effects when cells were treated with SB-204990. Similar results were obtained using the Ampk activator AICAR (Supplementary Fig. S7m–o). Importantly,

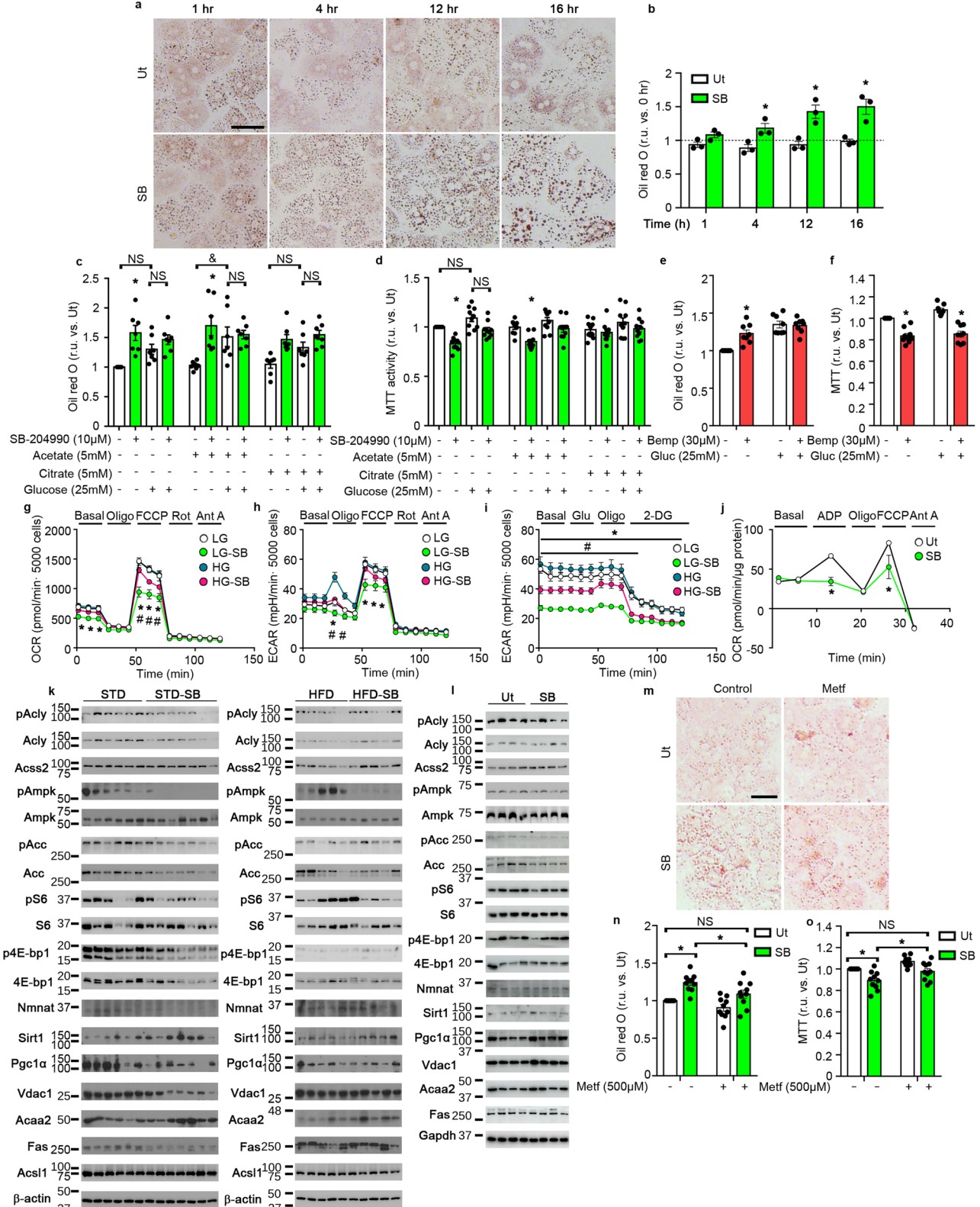

pharmacological inhibition and activation of other pathways altered by SB-204990, such as mTOR and folate cycle, did not revert alterations on MTT activity and lipid content promoted by SB-204990, indicating a major role of Ampk inhibition on these effects (Supplementary Fig. S7o). Together, these results indicate that SB-204990 modulates molecular mechanisms of aging, such as liponeogenesis and mitochondrial metabolic hubs, playing an important role in the control of energy metabolism via Ampk signaling.

## Discussion

The identification of efficient compounds able to reprogram cellular metabolism to prevent metabolic diseases is critical in the

**Fig. 7 Reduced Ampk signaling is required for SB-204990-mediated effects in mitochondrial function and lipid accumulation. a–j** Primary hepatocytes were treated with SB-204990 or bempedoic acid under different experimental conditions. Unless otherwise stated, hepatocytes were treated for 16 h with 10 μM SB-204990 or 30 μM bempedoic acid. **a** Representative images of primary hepatocytes stained with Oil red O treated with 10 μM SB-204990. Scale bar 50 μm. $n = 3$. **b** Quantification of Oil red O staining of primary hepatocytes treated with 10 μM SB-204990. $n = 3$. Two-way ANOVA. **c–d** Quantification of Oil red O staining and determination of metabolic activity by MTT test in primary hepatocytes treated with 10 μM SB-204990 in the presence of indicated doses of acetate or citrate. Glucose concentration in basal conditions (low glucose: LG) is 5 mM, and supplemented (high glucose: HG) is 25 mM. **c** Oil Red O staining. $n = 7$. Two-way ANOVA. **d** MTT test. $n = 10$. Two-way ANOVA. **e, f** Quantification of Oil red O staining and determination of metabolic activity by MTT test in primary hepatocytes treated with 30 μM bempedoic acid. Glucose concentration in basal conditions is 5 mM, and supplemented is 25 mM. **e** Oil Red O staining. $n = 8$. Two-way ANOVA. **f** MTT test. $n = 9$. Two-way ANOVA. **g** Oxygen consumption rate on primary hepatocytes. $n = 4$ for LG. $n = 5$ for LG-SB. $n = 5$ for HG. $n = 5$ for HG-SB. Two-way ANOVA. **h** Extracellular acidification rate on primary hepatocytes. $n = 4$ for LG. $n = 5$ for LG-SB. $n = 5$ for HG. $n = 5$ for HG-SB. Two-way ANOVA. **i** Extracellular acidification rate on primary hepatocytes on glycolysis stress test. $n = 5$ for LG. $n = 6$ for LG-SB. $n = 5$ for HG. $n = 4$ for HG-SB. Two-way ANOVA. **j** Oxygen consumption rate of freshly isolated liver mitochondria of mice gavaged with SB-204990. Liver samples were collected 3 h post-gavage. $n = 3$. Two-way ANOVA. **k** Immunoblots for hepatic proteins of mice treated with SB-204990 for 15 weeks. Densitometric quantification is shown in Supplementary Fig. S7f. $n = 6$ for STD. $n = 7$ for STD-SB. $n = 5$ for HFD. $n = 5$ for HFD-SB. **l** Immunoblots of primary hepatocytes treated or not with SB-204990 for 16 h. Densitometric quantification is shown in Supplementary Fig. S7i. $n = 4$. **m–o** Primary hepatocytes were treated with SB-204990 or not in the presence of metformin (500 μM) for 16 h. **m** Representative images of primary hepatocytes stained with Oil red O. Scale bar 50 μm. $n = 10$. **n** Quantification of Oil red O staining. $n = 10$. Two-way ANOVA. **o** Quantification of metabolic activity by MTT. $n = 10$. Two-way ANOVA. STD: standard diet. HFD: high-fat diet. SB: SB-204990. Bemp: Bempedoic acid. Glu: glucose. Oligo: Oligomycin. FCCP: Carbonyl cyanide-p-trifluoromethoxyphenylhydrazone. Rot: Rotenone. Ant A: Antimycin A. ADP: Adenosine diphosphate. 2-DG: 2-deoxyglucose. NS: not significant. r.u.: relative units. Ut: untreated. Data shown are the means ± SEM. Unless otherwise stated $*p < 0.05$ SB vs. Ut or LG-SB vs. LG. $\#p < 0.05$ HFD-SB vs. HFD. $\&p < 0.05$ HG-acetate vs. acetate.

---

current aging society. In humans, nutritional interventions promoting improved metabolic control, such as caloric restriction or exercise, are challenging or even not feasible given the medical condition of the individuals. Advances in the knowledge of molecular mechanisms of metabolic control have been obtained through studies using genetically modified organisms, limiting the translational potential of the results generated. In contrast, pharmacological interventions might, in principle, be easily applied in clinical practice. In this study, we show that SB-204990 improves metabolic and physical health in diet-induced obese mice, which are known to suffer premature death[37]. It is important to mention that SB-204990 elicits different physiological effects depending on the feeding diet. In mice fed with a healthy standard diet, SB-204990 produced a metabolic imbalance with moderate insulin resistance, which is reminiscent of the effects of the geroprotector mTOR inhibitor rapamycin as well as the lack of adipose Rictor[38–40], while SB-204990 produced improvements in glucoregulation in mice fed with HFD. In this regard, mice treated with SB-204990 fed with a healthy diet exhibited restricted phosphorylation of 4E-bp1, a well-known marker of mTOR activation, while another marker of mTOR activation, the phosphorylation of S6, was not significantly affected. These alterations in mTOR signaling were not detected in mice treated with SB-204990 fed with HFD. These results suggest that alterations in specific branches of mTOR signaling, which could be mediated by differential activation of phosphatases or other kinases that regulate targets of mTOR, might contribute to the development of compromised glucoregulation specifically in SB-204990-treated STD-fed mice. Whether compromised glucoregulation produced by SB-204990, similar to rapamycin, could produce unhealthy aging should be the focus of further research.

A potential molecular mechanism of cellular reprogramming that could occur under conditions of SB-204990-mediated Acly inhibition is the epigenetic modulation of histones via acetylation. Several reports have shown that restriction of Acly expression produces reduced levels of Ac-CoA and limited histone acetylation in cell cultures[23,41]. However, separate reports have also shown that upon Acly deficiency, acetylation levels of several histone residues are not altered in cell cultures and tissues[22,42]. Our results indicate that the acetylation of several histones can be enhanced by acetate but not by citrate, suggesting that

cytoplasmic Ac-CoA synthetase could play a greater role in providing Ac-CoA than Acly to produce global increases in histone acetylation. Under these conditions, the Acly inhibitor SB-204990 did not affect global protein acetylation levels, further suggesting that the role of Acly in this important cellular process is limited. The analysis of different tissues has shown that compensatory mechanisms might be involved in maintaining unaffected histone acetylation levels in vivo in mice fed with challenging diets or mice lacking Acly[22,43]. Among those, alterations in the expression or activity of Ac-CoA-generating enzymes, such as cytoplasmic Ac-CoA synthetase or malonyl-CoA decarboxylase, could participate in this process. In mouse liver, SB-204990 failed to modulate Ac-CoA levels and histone acetylation and non-histone protein acetylation levels, indicating that the effects of SB-204990 on hepatic cholesterol levels and overall physiology are separated from global changes in histone and non-histone protein acetylation. Remarkably, these data suggest that different availability of Ac-CoA, subcellular compartmentation of Ac-CoA, which has not been evaluated herein, or regulatory processes exist in the modulation of cholesterol synthesis and histone acetylation upon Acly inhibition[22]. Despite these results, it is not possible to rule out that transcriptional changes produced by SB-204990 are due to changes in histone acetylation levels in specific transcriptional regulatory regions of target genes. These potential changes could orchestrate metabolic arrangements that could contribute to the physiological effects of SB-204990.

Our multiomic analysis has identified novel mechanisms and confirmed the enrichment of well-established pathways that depend on Acly activity. Among expected pathways modulated under restricted Acly activity, cholesterogenesis appears to be greatly restricted. These data support the effectiveness of Acly inhibition as a therapeutic target to reduce cholesterol levels. In this regard, another Acly inhibitor that also has an Ampk activatory effect, bempedoic acid, has successfully undergone clinical trials as a cholesterol-lowering agent, and it is now approved by the FDA and the European Medicine Agency (EMA)[20]. Fatty acid metabolism is another anticipated pathway modulated in mice treated with SB-204990. Changes in fatty acid profiles have been observed in conditions of restricted Acly activity[31], and significant changes in phosphatidylserine levels and modulations on several phosphatidylcholines, diglycerides, and triglycerides

species were observed in our study. These changes, indicating a modulation towards longer phosphatidylcholine species in mice treated with SB-204990, might affect membrane fluidity, limiting the activity of proteins in lipid rafts that play an important role in cellular signaling, including insulin sensitivity[44]. New pathways altered by the Acly inhibitor SB-204990 include changes in amino acid metabolism, mTOR, folate cycle, and one-carbon metabolism. These pathways have been highlighted in genetic and nutritional models of extended longevity[27,45–47]. Remarkably, the control of mTOR on Acly activity and stability has been demonstrated[48], and previous observations indicated that the lack of Acly produces reduced mTOR signaling in differentiating adipocytes[49]. Our data indicate that targeting Acly in healthy feeding conditions produces reduced mTOR signaling in vivo, indicating the existence of effective crosstalk among master regulators of cellular metabolism and their targets. These central pathways play a role in controlling effectors of cellular energetics[50,51]. Multiomic analyses also highlighted effects on cellular energetics and mitochondria. Interestingly, SB-204990, as well as bempedoic acid, promoted lipid storage despite the fact that glucose incorporation into lipids was reduced in primary hepatocytes. Although similar results (e.g., larger lipid droplets/ adipocyte size) have been observed in other experimental models depleted of Acly activity[22,31–33], this is particularly interesting since Acly participates in de novo lipogenesis. Our results showing that SB-204990 and bempedoic acid produce net decreases in mitochondrial oxygen consumption suggest that reduced mitochondrial metabolism could contribute to accumulate lipids in hepatocytes treated with SB-204990 or bempedoic acid. Experiments performed in primary hepatocytes indicated that neither acetate nor citrate supplementation promoted lipid accumulation. These data suggest that restricted mitochondrial metabolism or alternative Ac-CoA sources, such as malonyl-CoA via malonyl-CoA decarboxylase activity, but not acetate or citrate, could contribute to accumulate intracellular lipids in hepatocytes treated with SB-204990 or bempedoic acid. Efficient mitochondrial performance is central to achieve longer and healthy life expectancy[52]. Effects of SB-204990 and bempedoic acid on mitochondrial function (i.e., producing restricted oxygen consumption) are similar to other pharmacological interventions that extend lifespan, such as rapamycin and metformin, which are considered mild mitochondrial poisons[36,51,53]. Despite lower oxygen consumption, mice exhibited lower activation of the master regulator of cellular energetics Ampk, which reflects a rich energetic status. The Ampk activators metformin and AICAR blunted the effects of SB-204990 in the alteration of metabolic activity and promotion of lipid storage while producing a synergistic effect in mitochondrial functional dynamics, suggesting a relevant role of Acly activity in cellular energetics. These effects were associated in vivo with restricted mTOR signaling and increased levels of Sirt1 (STD-SB mice) or the NAD$^+$ generator Nmnat (HFD-SB). Sirt1, Ampk, and mTOR have complex crosstalk but are undoubtedly intertwined in the control of Acly stability and play a central role in lifespan and healthspan[38,54,55].

Our study was carried out in a rather limited number of male C57BL/6 mice. Given the different responses to diets of male and female mice and sex-specific differences in metabolic control, sex differences are plausible and may be relevant[56–58]. Our work provides evidence of the effects in the hepatic tissue of the Acly inhibitor SB-204990. It remains to be investigated whether similar metabolic signatures occur in other organs. Several findings described herein must be taken into consideration in any potential further work using SB-204990 and await further investigation. In particular, the effects of Acly inhibitors on different feeding regimens, Acly-independent effects of SB-204990 and/or bempedoic acid, effects of acetate metabolism on longevity

pathways, and the possibility of targeting monocarbonated metabolism to achieve healthy aging in mammals represent central questions that deserve further research. Although further work is required, our study posits that pharmacological inhibitors of the Acly produce profound modulations on lipid and mitochondrial metabolism that depend on the diet consumed, which might have important consequences for physical health, metabolic control, and life expectancy.

## Methods

**Animals, diets, and in vivo treatments**. Mice experimentations were approved by the CABIMER Animal Committee and performed in accordance with the Spanish law on animal use RD 53/2013 and the EU Directive (2010/63/EU) for animal research. Eight-week old wild-type male C57BL/6 mice were purchased from Janvier Labs (Le Genest-Saint-Isle, France). Mice were housed in individually ventilated cages (Tecniplast, Buguggiate, Italy). Souralit plus 29/12 bedding (Souralit, Gerona, Spain) was sterilized by autoclave and added to each cage. Mice were group-housed at 3–4 mice per cage. Mice were maintained on a 12-h light/ dark cycle and had ad libitum access to rodent chow TD2914 (Envigo, Barcelona, Spain) and water until the initiation of the treatments. One cohort of mice was treated or not with SB-204990, mice were distributed in four different groups at 26 weeks of age, and treatments were initiated: (1) standard TD2914 diet (STD) with the following percentages of calories distribution: 67% from carbohydrates, 20% from proteins, and 13% from fats. (2) STD + 250 mg/kg of SB-204990 (STD-SB). (3) a cholesterol-free high-fat diet (HFD) TD.06414 (Envigo, Barcelona, Spain) with the following percentages of calories distribution: 21.4% from carbohydrates, 18.3% from proteins, and 60.3% from fats (37% saturated, 47% monounsaturated, and 16% polyunsaturated). (4) HFD + 250 mg/kg of SB-204990 (HFD-SB). At week 41 of life, these mice were euthanized at 16 h of fasting for the ex vivo analyses. Another cohort of mice was treated or not with SB-204990 starting at 5 weeks of age using the STD TD2914 diet. This cohort was subjected to indirect calorimetry analyses and was sacrificed at 10 weeks of life.

**Chemicals**. SB-204990 was purchased from Tocris and Sreeni Labs Private Limited, and bempedoic acid was purchased from Cayman Chemical.

**Metabolic tests**. For OGTTs, mice were fasted for 6 h at 10 a.m. and received a dose of glucose (3 g/kg) by gavage. For IPPTT, mice were fasted for 6 h from 10 a.m. and received an intraperitoneal injection of sodium pyruvate (2 g/kg). For the ITT, mice were fasted for 3 h from 10 a.m. and received an intraperitoneal injection of insulin (1.5 IU/kg). For the HOMA-IR, mice were fasted from 8 p.m., and determinations were performed at 16 h of fasting. To determine glucose levels, blood samples were taken by venipuncture using a Precision Xceed glucometer (Abbott, Madrid, Spain). Insulin was measured in plasma using ELISA kits (Crystal Chem, Downers Grove, IL, USA).

**RNA isolation and semi-quantitative RT–PCR**. Total RNA was isolated from frozen tissues using the easy-blue total RNA Extraction kit (#17061, iNtRON Biotechnology, Inc., Seongnam, Korea). RNA concentration and quality were determined with the NanoDrop® Spectrophotometer ND-100. Total RNA (0.5–2 μg) was used to synthesize cDNA with the iScript™ cDNA Synthesis Kit (Bio-Rad Laboratories). Primer sequences are presented in Supplementary Data 5. The mRNA expression was calculated by the $2^{-\Delta\Delta CT}$ method and normalized to the expression of Rps29.

**Acly activity**. Acly activity was determined as described in ref. [59]. In brief, liver tissue was lysed in ice-cold 220 mM mannitol, 70 mM sucrose, 5 mM potassium HEPES buffer, pH 7.5 containing 1 mM dithiothreitol. Lysates were centrifuged at $600 \times g$ for 10 min to precipitate the nuclei and debris. Then, the supernatant was centrifuged at $5500 \times g$ for 10 min to precipitate the mitochondrial fraction, and the supernatant was then centrifuged at $20,000 \times g$ for 20 min to generate a cytosolic fraction used to measure enzyme activity. Enzymatic activity was measured in 5 mM citrate, 0.3 mM coenzyme A, 3 mM ATP, 0.15 mM NADH, 10 mM MgCl$_2$, 10 mM dithiothreitol, and 6 units/ml of malate dehydrogenase in 100 mM Tris chloride buffer, pH 8.5 at 37 °C. The NADH disappearance was monitored at 340 nm for 1 min for background, after which 5 mM citrate was added to determine Acly activity.

**Primary hepatocyte isolation**. Primary hepatocytes were isolated from the livers of wild-type male mice as previously described with minor modifications[60]. In brief, livers were perfused with Hank's Balanced Salt Solution (5.33 mM KCl, 0.44 mM KH$_2$PO$_4$, 0.34 mM Na$_2$HPO$_4$, 138 mM NaCl, 4.17 mM NaHCO$_3$, and 25 mM HEPES) containing 5 mM glucose supplemented with 0.5 mM EGTA and 25 mM HEPES (pH 7.4 at 37 °C) using a CTP100 peristaltic pump (Thermo Fisher). Once the liver was perfused, the perfusion solution was changed to DMEM (Sigma-Aldrich D5546) supplemented with 100 U/ml Penicillin and 0.1 mg/ml

Streptomycin (pen-strep), 10 mM HEPES, and 100 U/ml of collagenase (Type IV, Worthington). Cells were released from the liver, and the cell suspension was filtered using a 70-μM cell strainer. Then, cells were centrifuged at $50 \times g$ for 2 min three times. Cell pellets were resuspended in DMEM (Sigma-Aldrich D5796) supplemented with pen-strep, 10 mM HEPES, 10 nM dexamethasone, 2 mM L-glutamine, 1 mM sodium pyruvate, and 10% fetal bovine serum (FBS) and seeded into plates precoated with collagen type I (Sigma-Aldrich St. Louis, MO, USA). Then, 60 min post-seeding the media was changed to DMEM (Sigma-Aldrich D5546) supplemented with pen-strep, 10 mM HEPES, 100 nM dexamethasone, and 10% FBS. Media was changed 3 h later to serum-free DMEM (Sigma-Aldrich D5546) supplemented with pen-strep, 5 mM HEPES, 10 nM dexamethasone, 2 mM L-glutamine, 1 mM sodium pyruvate, and cells were cultured for up to 16 h for various experiments.

**Pancreatic islet procuration and culture**. Pancreatic islets were isolated from wild-type mice by intraductal collagenase perfusion as previously described[61]. Islets were cultured for 16 h in RPMI 1640 supplemented with 10% FBS, pen-strep, 2 mM glutamine, 1 mM sodium pyruvate, 50 μM β-mercaptoethanol, and 10 mM HEPES.

**Glucose incorporation into lipids**. Glucose incorporation into lipids was determined in primary hepatocytes, as previously described, with minor modifications[44]. Primary hepatocytes were seeded in 12-well culture dishes at $2.5 \times 10^5$ cells per well and were incubated for 16 h under different experimental conditions. After a 60-min preincubation in 20 mM HEPES, pH 7.4, 114 mM NaCl, 4.7 mM KCl, 1.2 mM $KH_2PO_4$, 1.16 mM $MgSO_4$, 2.5 mM $CaCl_2$, 3 mM glucose, and 1% fatty acid-free bovine serum albumin at 37 °C, cells were treated with 30,000 cpm of [3-$^3$H] glucose (0.5 Ci/mol, Perkin Elmer NET331C005MC) and plates were sealed with an optical adhesive film. The incubation was terminated 2 h later by adding 1 ml of methanol:phosphate buffered saline (PBS) (2:3) to the cells. Cells were collected with gentle pipetting, centrifuged at $700 \times g$, and washed twice with PBS. Then, 200 μl of 0.2 M NaCl was added to the cell pellet, and the mixture was immediately frozen in liquid $N_2$. Samples were thawed, and a mixture composed of 750 μl of $CHCl_3$:methanol (2:1) and 50 μl of 0.1 N KOH per sample was added. Then, vigorous vortexing was applied, and the lipid and aqueous fractions were separated by centrifugation at $2000 \times g$ for 20 min. The top aqueous layer was discarded, and the bottom lipid-soluble layer was washed with 200 μl of methanol:water:$CHCl_3$ (48:47:3). Aliquots (200 μl) of the lipid-soluble phase were transferred into scintillation vials, and radiolabeled lipids were quantified using liquid scintillation cocktail (Perkin Elmer).

**Cell death**. Cell death was assessed by ELISA according to the instructions of the manufacturer (Sigma-Aldrich 115446775001). Optical density at 405 nm against 490 nm reference was determined with a Varioskan Flash spectrophotometer (Thermo Scientific). In the case of primary hepatocytes, cells were seeded in 12-well plates. In the case of pancreatic islets, 10 islets per well were used.

**MTT test**. MTT activity was determined using the Cell Proliferation Kit I according to the recommendations of the manufacturer (Roche, Spain). Optical density was determined at 575 nm with a reference wavelength of 690 nm using a Varioskan Flash spectrophotometer (Thermo Scientific, Spain). In the case of primary hepatocytes, cells were seeded in 12-well or 24-well plates. In the case of pancreatic islets, 35 islets per well were used.

**Urea test**. Urea production was determined in primary hepatocytes, as previously described, with minor modifications[62]. In brief, primary hepatocytes from wild-type mice were seeded in 12-well plates at a density of $3 \times 10^5$ cells per well and were incubated for 16 h under different experimental conditions. Then, media was replaced, and urea production was measured after 2 h using the MAK006 kit according to the manufacturer's instructions (Sigma-Aldrich St. Louis, MO, USA) using a Varioskan Flash spectrophotometer (Thermo Scientific).

**Glucose-stimulated insulin secretion**. Groups of 10 pancreatic islets were washed in 500 μl of Krebs-Ringer bicarbonate-HEPES buffer (KRBH) (140 mM NaCl, 3.6 mM KCl, 0.5 mM $NaH_2PO_4$, 0.5 mM $MgSO_4$, 1.5 mM $CaCl_2$, 2 mM $NaHCO_3$, 10 mM HEPES, and 0.1% BSA) and pre-incubated at 37 °C for 45 min in 300 μl of KRBH. Islets were then centrifuged, and the KRBH buffer was discarded. Subsequently, fresh KRBH supplemented with 2.2 mM glucose was added, and islets were incubated for 30 min. Next, the buffer was harvested (basal insulin secretion), and 500 μl of KRBH supplemented with 22 mM glucose was added. Islets were incubated for 30 min at 37 °C, and then the buffer was harvested (stimulated insulin secretion). Insulin levels were measured using a mouse insulin ELISA kit according to the manufacturer's instructions (Crystal Chem).

**Pharmacokinetic studies**. Wild-type mice received an oral gavage of SB-204990 (30 mg/kg of body weight), and serum samples were collected at different time points. SB-204990 in serum samples was quantified by a chromatography and mass spectrometry method. Briefly, 150 μl of acetonitrile containing

2,4,dichlorohydrocinnamic acid (internal standard) were added to 50 μl of serum samples, which were vortex-mixed and centrifuged at $20,000 \times g$ at 4 °C for 15 min. Then, 110 μl of supernatant was transferred to the autosampler vial for analysis. The chromatographic experiments were carried out on an Agilent 1290 HPLC system (Agilent, CA). The separation of SB-204990 was accomplished using a C18 Supelco $50 \times 2.1$ mm, 3 μm column set at 4 °C. The mobile phase consisted of water/acetonitrile 90/10 with 0.1% formic acid as component A and acetonitrile/water 90/10 with 0.1% formic acid as component B using a linear gradient. Mass spectrometry analysis was performed using the mass spectrometer model API 4000 system from Applied Biosystems/AB Sciex. The data were acquired and analyzed using Multiquant 2.1.1. (Sciex). Negative electrospray ionization data were acquired using multiple reaction monitoring (MRM). Standards were characterized using the MRM transitions (389/329) for SB-204990 and (218,9/174.8) 2,4,dichlorohydrocinnamic. The instrumental settings were 500 °C and 4500 V, and compound parameter settings for collision energy were −24eV for SB-204990 and −16eV for 2,4,dichlorohydrocinnamic.

**Indirect calorimetry**. Mouse metabolic rate was assessed by indirect calorimetry in an OxyletPro system (PanLab Harvard Apparatus). Mice were housed singly with water and food available ad libitum and maintained at ~22 °C under a 12:12-h light:dark cycle (light period 08:00–20:00). The concentrations of oxygen and carbon dioxide were monitored at the inlet and outlet of the sealed chambers to calculate oxygen consumption. Each chamber was measured for 60 s at 10-min intervals, and data were recorded for ~48 h total. Locomotor activity was monitored using an infrared photocell beam (rearing) and a sensor platform (activity). Food and water intake were automatically monitored using sensors in each cage.

**Barnes Maze**. The method was performed according to a previously published protocol[63]. In brief, on the pre-training trial, mice were pre-trained to enter the escape box, guiding them to the escape box. Mice remained in the box for 2 min. Then, training trials were initiated (4 days). Mice were trained four trials per day, and trials were separated by 15 min. Mice were allowed to explore the maze for up to 3 min and were guided to the escape box. Once the mice entered the box, the buzzer was turned off, and the mice remained in the box for 1 min. The following day, subjects received a probe trial for 90 s to determine short-retention memory. During the probe trial, the escape box was removed. Primary latency and total attempts were recorded. Without further training, 7 days after, mice were tested for another probe trial for 90 s to determine long-term retention memory.

**Fecal lipid content**. Lipid extraction from 1 g of dried feces was performed by chloroform-methanol as previously described in ref. [64]. In brief, feces were lysed, and 0.9% NaCl solution was added and vortexed. Then, a similar volume of chloroform:methanol (2:1) was added, and samples were vortexed. Samples were centrifuged at $1000 \times g$ for 10 min, and the lipid-containing nonpolar phase was extracted. Then, samples were air-dried, and lipids were quantified.

**Glycated hemoglobin**. HbA1c levels were determined in blood samples according to the manufacturer's protocol (Crystal Chem).

**Cholesterol and LDL/VLDL determinations**. Cholesterol and low-density/very low-density lipoprotein (LDL/VLDL) in serum were determined using the Enzy-Chrom Assay kit (BioAssay Systems, E2HL-100), according to the manufacturer's instructions.

**Rotarod**. Results from rotarod tests are presented as the time to fall from an accelerating rotarod (4–40 rpm over 5 min). Mice were given a 1-minute habituation trial at 4 rpm on the day before the experiment. The results shown are the averages of three trials per mouse[65].

**Wire hang**. For the wire hang test, mice were allowed to grip a horizontal 1 mm wire with four paws for up to 60 s, and the latency to fall from the wire was determined. Three different trials were performed with each mouse, and the results shown are the averages of the three trials[65].

**Determination of AST and ALT**. Serum concentrations of AST and ALT were measured using a Cobas Integra 400 plus automated analyzer (Roche Diagnostics).

**Histochemistry and cytochemistry**. Dissected tissues and cell cultures were fixed in 4% paraformaldehyde. Tissue sections (5 μm thick) were deparaffinized and rehydrated as previously described in ref. [60]. For hematoxylin and eosin staining, sections were immersed in hematoxylin (Merck) and eosin (Merck) for 4 and 2 min, respectively. Evaluation of lipid content was performed using Oil red O staining. Quantification of lipid content was performed after isopropanol solubilization at 510 nm. The histological study of the stained sections was carried out using a Leica DM6000B microscope equipped with a DFC390 camera (Leica, Barcelona, Spain) and an Olympus IX71 with an Olympus DP70 camera

(Olympus). Adipocyte size was quantified with ImageJ software (National Institutes of Health).

**AML12 cell culture**. AML12 cells were obtained from ATCC and were maintained and propagated in DMEM/nutrient mixture F-12 Ham with 10% fetal bovine serum, pen-strep, ITS Liquid Media Supplement (Sigma-Aldrich), and 0.1 μM dexamethasone. For experiments, cells were exposed to 10% dialyzed FBS in DMEM/nutrient mixture F-12 Ham containing 17 mM glucose and were treated with additional glucose (up to 42 mM), citrate, acetate, and SB-204990 for 16 h. Samples were snap frozen and maintained at $-80\,°C$ until processed.

**Acid histone extraction**. AML12 cell pellets ($2 \times 10^6$ cells) and liver samples were cut into small pieces (60–70 mg) and transferred to an Eppendorf tube. Thereafter, 1 ml of Triton Extraction Buffer (TEB; PBS containing 0.5% Triton X 100 (v/v), 0.02% (w/v) $NaN_3$) + inhibitors (Trichostatin 10 μM, nicotinamide 10 mM, sodium butyrate 50 mM, with protease, and phosphatase inhibitors P0044, P5725, and P8340) were added per 200 mg of tissue/$1 \times 10^7$ cells, and samples were homogenized. Then, lysates were incubated on rotation for 10 min at 4 °C, and the mixture was centrifuged at 2000 rpm for 10 min at 4 °C. The supernatant was removed, and the pellet was resuspended in 100 μl of 0.2 N HCl and incubated on rotation overnight at 4 °C. Afterwards, samples were centrifuged at 2000 rpm for 10 min at 4 °C, and the supernatant was transferred to a new tube. Next, HCl was neutralized with NaOH 0.2 N, and samples were stored at $-80\,°C$ until used.

**Western blot**. For total protein isolation, samples were lysed in radio-immunoprecipitation assay buffer (20 mM Tris–HCl, 150 mM NaCl, 1 mM $Na_2$EDTA, 1 mM EGTA, 1% NP-40, 1% sodium deoxycholate, pH 7.5) with protease, phosphatase, and deacetylase inhibitors P0044, P5725, P8340, and SC-362323. Western blots were performed according to standard methods, which involved, in certain cases, membrane stripping and incubation with a primary antibody of interest, followed by incubation with a horseradish peroxidase-conjugated secondary antibody and enhanced chemiluminescence. Antibodies used are presented in Supplementary Data 5. Blots were quantified with ImageJ, and the bands of interest were normalized to Ponceau S, β-actin, and/or Gapdh staining, as previously validated in refs. [66,67].

**Ac-CoA determination**. Ac-CoA was determined using the MAK039 kit following the instructions of the manufacturer (Sigma-Aldrich).

**Transcriptome profiling**. cRNA preparations were hybridized to mouse Clariom™ S Assay Array chips using the standard protocols of the Genomic Core Facility of CABIMER (Affymetrix, Santa Clara, CA, USA). Briefly, 50 ng of total RNA from each biological sample was used as a template to prepare biotinylated fragmented cRNA using a two-cycle amplification step. Image analysis, quality control, and quantification of data were performed using the Affymetrix GeneChip Command Console Software 2.0. Statistic data analysis was performed using the Transcriptome Analysis Console (TAC), which included the processing of fluorescence data (raw data), data normalization, the exchange value of expression of a condition regarding control, and statistical parameters appropriate to establish a degree of credibility (p-value) (Affymetrix). Transcripts that showed p-value <0.05 were selected as differentially expressed genes. In the TAC analyses module, Wikipathways was used. Raw data are accessible at GSE196853.

**Untargeted metabolomics**. Metabolite extraction was performed in frozen tissues using 400 μl of MeOH:water (8/1, v-v). After vortexing, samples underwent five repeated freeze–thaw cycles and were kept on ice for 1 h. Then, samples were centrifuged (10 min, 15200 rpm at 4 °C), and 100 μl and 250 μl of supernatant were transferred to autosampler vials for respective liquid chromatography/mass spectrometry (LC/MS) and gas chromatography/mass spectrometry (GC/MS) analysis. For LC/MS and MS/MS analysis, samples were injected into a UHPLC system (1290 Agilent) coupled to a quadrupole time of flight (QTOF) mass spectrometer (6550 Agilent Technologies) operated in positive electrospray (ESI+) and negative electrospray (ESI−) ionization mode. Tissue extracts were separated using a Waters Acquity UPLC BEH HILIC (1.7 μm, 2,1 × 150 mm). The solvent system was A = 50 mM ammonium acetate in water and B = acetonitrile/water with 50 mM ammonium acetate. The linear gradient elution used started at 95% A (time 0–2 min) and finished at 95% B (9.5 min). The injection volume was 2 μl. ESI conditions were gas temperature 150 °C, drying gas 11 L/min, nebulizer 35 psig, fragmentor 120 V, and skimmer 65 V. The instrument was set to acquire over the m/z range of 50–1200 with an acquisition rate of 3 spectra/s. MS/MS was performed in targeted mode, with a default iso width (the width half-maximum of the quadrupole mass bandpass used during MS/MS precursor isolation) of 1,3 m/z (narrow). The collision energy was fixed at 10, 20, 30, and 40 V. For GC/MS analysis, samples were dried under a stream of $N_2$ gas and lyophilized before chemical derivatization with 40 μl of methoxyamine in pyridine (30 μg/ml) for 45 min at 60 °C. Subsequently, samples were silylated using 25 μl of N-methyl-N-trimethylsilyltrifluoroacetamide with 1% trimethylchlorosilane (Thermo Fisher Scientific) for 30 min at 60 °C to increase the volatility of metabolites. A 7890B GC

system coupled to a 7250 QTOF mass spectrometer (Agilent Technologies) was used for GC-MS analysis. Derivatized samples (1 μl) were injected in the gas chromatograph system equipped with an Agilent 19091S-433UI HP5-ms Ultra Inert 86 stationary phase column (30 m × 0.25 mm × 0.25 μm). Helium was used as a carrier gas at a flow rate of 1.5 ml/minute in constant flow mode. The temperature gradient used was from 70 to 190 °C at a heating rate of 11 °C/min and from 190 to 325 °C at 21 °C/minute, finally holding for 4 min. Electron impact ionization was conducted at 70 eV, and the source temperature was set to 230 °C. Mass spectra were recorded after a solvent delay of 3 min, with the analyzer acquiring in full-scan MS mode at a rate of 5 scan/sec within a mass range of 35–700 m/z. Upon acquisition, raw LC/MS proprietary vendor files were converted to the open standard format *mzML* using Proteowizard MS-convert[68] and subsequently processed by XCMS software (version 3.10.2)[69]. XCMS analysis of these data provided a matrix containing the retention time, m/z value, and integrated peak area for aligned features across samples (a feature being defined as molecular entity with a unique m/z and a specific retention time). QC samples were used to account for analytical variation. In LC/MS experiments, features significantly modified by SB-204990 in STD and HFD were prioritized for MS/MS identification in targeted mode. Metabolites were identified as conforming to Level 2, as specified by Schymanski et al., since their accurate mass and experimental MS/MS spectra coincide with the fragmentation pattern of chemical standards from the METLIN, MassBank, and/or NIST17 databases[70]. For GC/MS analyses, raw GC/MS data were converted to *mzXML* format using Proteowizard MS-convert. Subsequent data deconvolution and alignment were performed using eRah, and metabolite identification was conducted by matching fragment spectra with compound spectra present in the Golm database and NIST17 libraries[71]. Data processing was conducted in R versions 3.6.1 and 4.0.3 (R-Foundation for statistical computing, www.Rproject.org). Raw data are accessible at 10.5281/zenodo.6222260.

**Proteomic analysis**. iTRAQ labeling with two 8-plex iTRAQ kits was performed in the Proteomic facility of the Institute of Biomedicine of Seville using their standard protocols. Individual liver samples were isolated in urea lysis buffer (8 M urea, 25 mM Tris, 100 mM NaCl, 25 mM NaF, 10 mM $Na_4P_2O_7$, 50 mM β-gly-cerophospate, 1 mM $Na_3VO_4$, 1:100 protease inhibitors, and 1:100 deacetylase inhibitors, pH 8). Samples were sonicated for 10 s and centrifuged at $20,000 \times g$ for 15 min at 4 °C. The supernatant was harvested and stored at −80 °C. Then, samples were reduced with 50 mM tris-(2-carboxyethyl)phosphine (AB Sciex) at 60 °C for 1 h with shaking and were subsequently alkylated using 200 mM S-methyl methanethiosulfonate (AB Sciex) for 30 min at room temperature. Samples were then trypsinized at 37 °C in a 10:1 ratio (w/w) of substrate/enzyme in a water bath overnight (Promega). Then, samples were speed-vac dried. The iTRAQ-labeling assay was conducted according to the manufacturer's instructions (iTRAQ 8-plex, AB Sciex). Briefly, peptides were reconstituted in 1 M triethylammonium bicarbonate (Sigma-Aldrich St. Louis, MO, USA), 0.05% SDS, 1:100 phosphatase inhibitor cocktail, 1:100 protease inhibitor cocktail, and 0.002% benzonase (Novagen, Argentina) and labeled with one isobaric amine-reactive tag. After 2 h of incubation, labeled samples were pooled, dried at 45 °C, and stored overnight at 4 °C. iTRAQ-labeled samples were desalted using Oasis HLB C18 cartridges (Waters, Milford, MA, USA) and dried using a vacuum centrifuge. Peptides were then prefractionated using MCX Oasis columns (Waters) and increasing concentrations (50–2000 mM) of ammonium formate. Fractions were collected, individually washed using C18 cartridges, and dried. Peptides from each fraction were separated using nano-liquid chromatography (nano LC 1000, Thermo Scientific) and analyzed by means of nano-electrospray ionization (Proxeon Biosystems, Odense, Denmark) connected to a Q Exactive Plus Orbitrap mass spectrometer (Thermo Scientific). Briefly, 13 μl of each fraction was loaded, preconcentrated, and washed in an Acclaim PepMap (75 μm × 2 cm, nanoViper, C18, 3 μm, 100 Å) precolumn (Thermo Scientific). Peptides were separated in an analytical column (75 μm × 15 cm, nanoViper, C18, 2 μm, 100 Å, Acclaim PepMap RSLC) for 240 min at 300 nL/minute (Thermo Scientific). Peptides were eluted with a gradient of buffer A (0.1% formic acid, 100% $H_2O$) to buffer B (0.1% formic acid, 100% acetonitrile). The Q Exactive system was used for MS/MS analysis in the positive ion and information-dependent acquisition mode. Proteins were identified and quantified using Proteome Discoverer (v2.1, Thermo Fisher Scientific), using three embedded search nodes; Mascot[72] (v2.5.1), Sequest HT (Thermo Fisher Scientific), and MS Amanda[73] (v2.1.5) search algorithms. The Percolator algorithm was used to calculate the false discovery rate (FDR) of peptide spectrum matches, set to a q-value of 0.05[74]. Entrez labels and gene names were retrieved with the R interface to Uniprot web services R package version 2.20.0. Raw counts were normalized using the weighted trimmed mean of M-values method[75], and the batch effect for the two iTRAQ experiments was removed using ComBat[76]. The contrast between different conditions was carried out with the quasi-likelihood test implemented in edgeR[77]. Raw data are accessible at 10.5281/zenodo.6140992.

**Multiomic analyses**. Transcriptomic, proteomic, and metabolite profiles were analyzed using MetaboAnalyst version 5.0[26] and Ingenuity pathway analysis (IPA) (Qiagen). Significantly modulated transcripts, proteins, and metabolites were included in the analyses, and protein levels were prioritized when protein and transcript levels were available. Multiomics analyses were performed utilizing the Joint Pathway Analysis (JPA) module from MetaboAnalyst 5.0. including the

settings "All pathways; hypergeometric test, closeness centrality, combine queries". IPA analyses used the modules Upstream Regulators, ToxList, and Canonical Pathways.

**Hepatic lipid analyses**. Hepatic lipids were extracted from ~20 mg of tissue using hexane-isopropanol, as previously described in ref. [64]. Samples were analyzed by GLC as previously described in ref. [78]. In brief, fatty acid methyl esters (FAMES) were obtained from isolated lipids by heating the samples at 80 °C for 1 h in 3 ml of methanol/toluene/$H_2SO_4$ (88:10:2 v/v). Heptadecanoic acid (1/10 w/v) was added to each sample as an internal standard to allow quantification. After cooling, 1 ml of heptane was added, and the samples were mixed. FAMES were recovered from the upper phase and then separated and quantified using a Hewlett–Packard 5890A gas chromatograph (Palo Alto, CA, USA) with a Supelco SP-2380 capillary column of fused silica (30 m length, 0.25 mm i.d., 0.20 μm film thickness) (Bellefonte, PA, USA). Hydrogen was used as the carrier gas, with a linear gas rate of 28 cm/s. The detector and injector temperatures were set at 220 °C, and the oven was set at 170 °C, with a split ratio of 1:50. Fatty acids were identified using standards (Sigma-Aldrich, St. Louis, MO, USA). Triacylglycerides and cholesterol esters were separated and quantified by GLC as previously described[79] with an Agilent 6890 gas chromatograph (Palo Alto, CA, USA). Triheptadecanoic acid was added to the samples as the internal standard for quantification. The injector and detector temperatures both were 380 °C, the oven temperature was 345 °C, and a head pressure gradient from 70 to 120 kPa was applied, changing this last parameter depending on the column. The gas chromatography capillary column was a J & W Scientific DB-17HT (15 m length, 0.25 mm i.d., 0.15 μm film thickness) (Folsom, CA, USA), with a linear gas rate of 50 cm/s, the split ratio was 1:80, and the detector was a flame ionization detector (FID). The different triacylglycerides and cholesterol esters were identified with respect to known samples, and the FID response was corrected. Polar lipids were analyzed and quantified by HPLC[80]. Separation by HPLC was carried out in a Waters 2695 Module (Milford, MA) equipped with a Waters 2420 ELSD. The column used was a Lichrospher 100 Diol 254-4 (5 μm; Merck) applying a method based on a linear binary gradient of solvent mixtures containing different proportions of hexane, 2-propanol, acetic acid, water, and trimethylamine. The flow was 1 ml/min, data were processed using Empower software, and the ELSD was regularly calibrated using commercial high-purity standards for each lipid.

**Oxygen consumption and ECAR**. Mitochondrial bioenergetics on primary hepatocytes were measured using an XF24 Extracellular Flux Analyzer (Agilent)[81]. After 16 h of treatment, cells were washed with Seahorse assay media (Seahorse Bioscience), which was supplemented with glucose at 10 mM, pyruvate at 1 mM, and glutamine at 2 mM, pH 7.2. Cells were incubated in a $CO_2$-free incubator at 37 °C for 1 h. Then, OCR and ECAR were measured. OCR and ECAR were determined in basal conditions and through consecutive injections of oligomycin (4 μM) at minute 27, carbonyl cyanide 4-(trifluoromethoxy) phenylhydrazone (FCCP; 2 μM) at minute 52, rotenone (1 μM) at minute 78, and antimycin A (5 μM) at minute 104. For the glycolysis stress test, after 16 h of treatment, cells were washed with Seahorse assay media (Seahorse Bioscience), which was supplemented with L-glutamine at 2 mM, pH 7.2. Cells were incubated in a $CO_2$-free incubator at 37 °C for 1 h. ECAR was determined in basal conditions and through consecutive injections of glucose (10 mM) at minute 27, oligomycin (1 μM) at minute 52, and 2-deoxyglucose (50 mM) at minute 78. OCR rates in fresh liver mitochondria were determined in samples of mice exposed to an oral gavage of SB-204990 (30 mg/kg of body weight) following the isolated mitochondria protocol of Agilent. In brief, 3 h after gavage, the liver was extracted, and the quadrate lobule was transferred to a gentleMACS™ dissociation tube with 5 ml of ice-cold MSHE buffer (70 mM sucrose, 210 mM mannitol, 5.0 mM HEPES, 1.0 mM EGTA, 0.5% (w/v) fatty acid-free BSA, pH 7.2). Subsequently, the tissue was homogenized on a gentleMACS tissue dissociator under the mouse-mito tissue isolation cycle protocol. Fresh samples were filtered twice through a 40-μm mesh filter and finally through a 10-μm mesh filter. The filtrate was centrifuged at 9000 × g for 10 min at 4 °C, and the pellet was resuspended in 200 μl of ice-cold BSA-free MSHE to quantify protein content. Then, fatty acid-free BSA 0.5% (w/v) was added, and samples were diluted in warm (37 °C) MAS buffer (70 mM sucrose, 220 mM mannitol, 10 mM $KH_2PO_4$, 5 mM $MgCl_2$, 2 mM HEPES, 1 mM EGTA, 10 mM succinate, 2 μM rotenone, 0.5% (w/v) fatty acid-free BSA, pH 7.2). Then, 5 μg of mitochondria per well were plated in Seahorse plates, and plates were centrifuged at 2000 × g for 20 min at 4 °C using a swinging bucket microplate adapter. OCR was determined through consecutive injections of ADP (4 mM) at minute 13, oligomycin (3.16 μM) at minute 20, FCCP (4 μM) at minute 26, and antimycin A (4 μM) at minute 31.

**Statistics and reproducibility**. The statistical analysis was performed using SigmaPlot 12.0 (SigmaPlot, Barcelona, Spain) and GraphPad Prism 7 (GraphPad Software Inc, San Diego, CA). Different statistical tests based on the number of groups and recommendations of the statistical SigmaPlot 12.0 and GraphPad Prism 7 were used. Statistical tests are reported in figure legends. Normality was assumed in statistics. Dunn's post hoc test was used in ANOVA on ranks. Dunnett's post hoc test was used in Supplementary Fig. S2c, and Bonferroni post hoc test was used

in Fig. 7b and Supplementary Fig. S2g, i, m, o. The remaining analyses were performed using Tukey's post hoc test. Data are shown as means ± SEM. Significance is reported at $p \leq 0.05$.

**Reporting summary**. Further information on research design is available in the Nature Portfolio Reporting Summary linked to this article.

## Data availability

Further information and request for resources and regents should be directed to and will be fulfilled by the lead contact, A.M.M. (alejandro.martinmontalvo@cabimer.es). A list of the main resources is included in Supplementary Data 5. Unedited blots are shown in Supplementary Fig. S8. Source data underlying the graphs and charts in the main manuscript can be found at https://doi.org/10.6084/m9.figshare.21912012.v1. Accession ID for trancriptomic data: Gene Expression Omnibus GSE196853 Accession ID for metabolomic data: Zenodo https://doi.org/10.5281/zenodo.6222260. Accession ID for proteomic data: Zenodo https://doi.org/10.5281/zenodo.6140992.

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

## Acknowledgements

This work was funded by grants from the Ministerio de Economía y Competitividad, Instituto de Salud Carlos III, co-funded by Fondos FEDER (PI15/00134, PI18/01590, CPII19/00023 to A.M.M.) and the Ministerio de Ciencia e Innovación (PID2021-123965OB-100 to A.M.M.). A.M.M. is funded by the Junta de Andalucía P20_00480, the Spanish Society of Diabetes, and CSIC 202220I059. M.S.K. is funded by the Nordea Foundation (#02-2017-1749), the Novo Nordisk Foundation (#NNF17OC0027812), the Neye Foundation, the Lundbeck Foundation (#R324-2019-1492), the Ministry of Higher Education and Science of Denmark (#0238-00003B). V.C.G. is funded by the Instituto de Salud Carlos III (CP19/00046), co-funded by FEDER. F.M. is funded by the CIBERDEM of the Instituto de Salud Carlos III. A.M.M. is the guarantor of this work and, as such, had full access to all the data in the study and takes responsibility for the integrity of the data and the accuracy of the data analysis. We acknowledge the support of the group of basic research on diabetes of the Spanish Society of Diabetes.

## Author contributions

A.S.G., M.A.C.M., I.E., C.P.M., D.G.M., G.M.C., R.L.F.S., M.E.M.V., A.J.N.P., L.L.N., M.P., L.M.C.M., M.S.K., and A.M.M. performed most of the experiments. E.M.F. performed lipid analyses. O.Y. and M.V. performed metabolomic analyses. D.L.L. and J.D. performed omic analyses. J.C.R., F.M., B.R.G., M.S.K., V.C.G., and A.M.M. interpreted the data. A.M.M. conceptualized the study, secured funding, and wrote the manuscript. All authors discussed the results and commented on the manuscript.

## Competing interests

The authors declare the following competing interests: Patent details: Inhibidores de la ATP-citrato liasa como agentes geroprotectores, P202230801, V.C.G., A.M.M, Spain. All other authors declare no conflict of interest.
