## [Peer Review File · Communications Biology]

Reviewers' comments:

Reviewer #1 (Remarks to the Author):

This manuscript mainly finds that long-term exposure to the Acly inhibitor SB-204990 can extend across evolutionary boundaries and improve changes in aging-related glucose and lipid metabolism indicators, further comprehensively carried out multi-omics integration transcriptomic, proteomic and untargeted metabolomic analysis. The data are sufficient, the content is comprehensive, the topic is novel, and the innovation is strong. However, the writing and the presentation of the results are relatively rough, and it is recommended to accept it after major revision.

Major comments:

1. The pictures are not standardized enough: the presentation of many pictures is too simple, lack of legends and abscissas; it is difficult for people to view, please add relevant content and modify;
2. The author introduces related research from aging, which is relatively new. And it was found in the study that SB-204990 would extend lifespan and health span across evolutionary boundaries, but the latter study only involved metabolic abnormalities related to aging, so have the authors directly by looked at the related indicators of aging in the 4 groups of mouse treatment models?
3. Regarding result 2, the author should summarize and improve the group differences in glucose metabolism-related indicators that appear in this section, and discuss them further, so as to make the setting of standard diets and high-fat diets more meaningful;
4. In terms of compound selectivity, can the same findings be reproduced in FDA-approved ACLY inhibitor?

Minor comments:

1. The reference format is confusing, and it is recommended to refer to the journal for revision;
2. The language description is obscure. It is recommended to modify and polish ;

Reviewer #2 (Remarks to the Author):

In this study, Alejandro Sola-García et al showed the effects of Acly inhibition on aging. They first demonstrated that the restriction of Acly could refrain intrinsic aging processes, as well as in flies and yeast, further indicating a conserved roles. In molecular level, they showed an independent role of Acly inhibition on histone acetylation, and some central metabolic pathway, as well as AMPK signaling were altered upon SB204990 treatment. They also showed Acly inhibition negatively affects the mitochondrial function, as indicated with altered OCR and ECAR. I have some recommendations might help the authors to improve their work.

1. I don't think Acly inhibition makes an effective role in physiological aging, as the authors also use HFD mice as the experimental materials. So I suggest the authors should also mention high glucose metabolism in the title and they are suggested not to overclaim their conclusion.
2. For the independent role of histone acetylation, the authors should explain in the discussion, since it is quite different as previously reported.
3. It will be good to verify the main molecular pathways identified by the authors (AMPK signaling pathway), maybe with some experiments in cell models.

4. The authors should improve the quality of the figures and adjust the color if possible.

Reviewer #3 (Remarks to the Author):

In the submitted manuscript, the authors investigate the downstream consequence of ATP citrate lyase (ACLY) inhibition. To this end, they employ SB-204990, a widely used cell permeable gamma-lactone prodrug with a well-defined pharmacokinetics and biological modulatory activity on lipid and cholesterol biosynthesis. The authors initially explore the impact of SB-204990 on chronological aging of *Saccharomyces cerevisiae* as well as on *Drosophila melanogaster* survival and locomotory activity. Then, they report the metabolic consequence of SB-204990 treatment on wild type C57BL/6 males. By performing metabolomics, transcriptomics and proteomics, the authors delineate the metabolic changes associated with SB-204990 treatment. The authors claim that their findings reveal novel molecular pathways linked to aging, which may be relevant to better understand metabolic diseases.

Overall, the manuscript requires a substantial revision. The submitted data are not well connected and sometime their biological meanings suffer of misinterpretation. While the effect of SB-204990 on mouse metabolism could be of interest, other aspects are inconclusive and in part redundant with previously published findings. Moreover, the relevance of these data on aging and health span is poorly supported. In my opinion, the manuscript requires a substantial work to be considered as a good candidate for Communications Biology.

Major points of concerns and comments:

1. In Fig.1, the authors describe glucose and insulin levels in young vs old mice. Moreover, they report expression changes in ACLY in mouse tissues. From these datasets, they test the impact of ACLY inhibition on longevity pathways using yeast and flies. As a reader, the two parts are completely unrelated. To better connect these multiple datasets, the authors should test the effect of SB-204990 on body weight, glucose tolerance and insulin levels in young vs old mice. This would give a rationale to run some experiments in yeast and *Drosophila melanogaster*.
2. The authors test SB-204990 in *Saccharomyces cerevisiae* strain BY4741. To the best of my knowledge, ATP-citrate lyase is not expressed in non-oleaginous yeast (Boulton C.A. et al, 1981; Rodriguez S. et al, 2016; Baroni M.D. et al, 2020). If so, SB-204990 has an off-target effect in Fig. 10. Or did the author use an ACL/ACLY expressing yeast strain? Please, clarify this issue.
3. In Fig. 1P-Q, the authors employ *Drosophila melanogaster* and explore SB-204990 effects on survival and locomotory activity. Based on their submitted data, SB-204990 extends the median lifespan of a few days. Since the authors assessed only 30 animals (n=1?), a more robust statistic would require additional biological replicates. Moreover, the authors should include ACLY mutant flies and treat them with saline (or DMSO?) and SB-204990 to rule out any off-targets. As an additional remark on Figure 1, it seems that SB-204990 may influence locomotory activity only in young animals. If so, SB-204990 has little impact on age-related processes, which would undermine some of the authors' conclusions.
4. I am puzzled by Fig. 3. It seems clear that SB-204990 has a negative impact on lipid metabolism in animals kept under standard diet. Thus, SB-204990 would not be a recommended treatment to maintain healthy lifespan in the majority of non-obese people, thereby undermining all the line of arguments supporting the use of ACLY inhibitors to promote longevity.
5. The authors claim that SB-204990 may influence physical performance in mice under HFD, which is expected if mice weight less. Surprisingly, HFD does not cause spatial memory deficits (Fig. S3), which is against what largely reported in literature. It would be recommended to review the experimental protocols and then draw the conclusions. In addition, the authors should include some histological analysis of brains from HFD mice, treated or untreated with SB-204990. Here, the authors should determine, whether SB-204990 can reduce neuroinflammation due to HFD.
6. The authors claim that ACLY inhibition is not associated with global modulation of histone acetylation. While the western blots may be conclusive and supporting the authors' arguments, it is

not possible to rule out that transcriptional changes are due to histone posttranslational acetylation at target genes.

7. Seahorse experiments: were the traces normalized to the number of cells or ug protein (as in Fig. 7G)? The authors wrote that "maximal OCR was diminished in cells cultured on high glucose (25 mM) treated with SB-204990". Fig. 7E shows a decreased basal respiration and maximal response in LG-SB (empty circle) compared to LG (full circle). Based on figure legend S2, it is reported that Lg: low glucose, Hg: high glucose. Could the authors clarify this discrepancy? Regarding the ECAR, I would strongly advice the authors to review their data. It may be helpful to use the glycolysis stress test to have a better picture of glycolysis in their cultured cells.

8. In Fig. 7, the authors report a large number of immunoblots for various kinases and their downstream targets. While I appreciate the authors' effort to dissect ACLY-dependent signaling pathway, it is very difficult to follow their line of arguments. The data are confusing and it is not possible to understand how ACLY influences AMPK and mTOR signaling. It would be important to re-organize these large datasets in a more understandable manner. Moreover, I would recommend to employ genetic approaches (e.g., shRNA) to mechanistically support their conclusions.

9. The authors claim that metformin and AICAR partially rescue lipid content promoted by SB-204990. Do they rescue also the mitochondrial defects?

Dear Dr. Rogers,

We are pleased to resubmit our work entitled "**Metabolic reprogramming by Acly inhibition using SB-204990 alters gluco-regulation and modulates molecular mechanisms of aging**" to Communications Biology. We have addressed all comments from the three reviewers. We include a point by point response marked in italic for all comments provided:

Reviewer #1 (Remarks to the Author):

This manuscript mainly finds that long-term exposure to the Acly inhibitor SB-204990 can extend across evolutionary boundaries and improve changes in aging-related glucose and lipid metabolism indicators, further comprehensively carried out multi-omics integration transcriptomic, proteomic and untargeted metabolomic analysis. The data are sufficient, the content is comprehensive, the topic is novel, and the innovation is strong. However, the writing and the presentation of the results are relatively rough, and it is recommended to accept it after major revision.

Major comments:

1. The pictures are not standardized enough: the presentation of many pictures is too simple, lack of legends and abscissas; it is difficult for people to view, please add relevant content and modify;

We would like to thank the reviewer for her/his positive comments, which have helped us to improve the presentation of our work. We have changed all figures in the manuscript to include dot plots with a more intuitive color code. Moreover, specific legends and all pertinent information in abscissas have been included to clarify the experimental groups and the determinations shown in this work.

2. The author introduces related research from aging, which is relatively new. And it was found in the study that SB-204990 would extend lifespan and health span across evolutionary boundaries, but the latter study only involved metabolic abnormalities related to aging, so have the authors directly by looked at the related indicators of aging in the 4 groups of mouse treatment models?

As highlighted by the reviewer, we show that SB-204990 extended lifespan in yeast and flies, showing increased spontaneous mobility early in the lifespan of flies. In mice, we determined metabolic health and we included other determinations of healthspan such as locomotor function (wire hang and rotarod) and neurocognitive health (Barnes Maze test for spatial memory) (Figure 3 and Figure S3). In these experiments we determined that locomotor function was improved in HFD-SB-204990-treated mice when compared to HFD-fed mice. There were no significant effects in locomotor function when STD-SB were compared with STD mice. Experiments related to neurocognitive health did not show significant differences in any experimental group. Based on the comments of another reviewer, we have evaluated in the brain

protein expression levels of markers of neuroinflammation, such as *Ccl2* (Figure S3G) (see specific response to reviewer

3 for a detailed description of the data). We have determined that HFD-fed mice exhibit increased *Ccl2* levels in the brain when compared to the brain of mice fed with healthy STD, while other markers of neuroinflammation were unaffected. Mice treated with SB-204990 fed with a HFD exhibited significant lower *Ccl2* levels in the brain when compared to HFD mice suggesting that, to some extent, SB-204990 reduced HFD-induced neuroinflammation. Following the advice of the reviewer, we considered interesting to monitor spontaneous activity of young mice treated with SB-204990, since flies exhibited increased spontaneous activity early on during the longevity assay. We have treated young mice with SB-204990 for 5 weeks. We have conducted experiments in metabolic cages (indirect calorimetry system) that monitor the activity of the mice (activity and rearing) while metabolic determinations are evaluated. Our results indicate that neither metabolic determinations (energy expenditure, respiratory quotient, energy intake and water intake) nor spontaneous activity (rearing and activity) were altered in young mice treated with SB-204990. We have described these new data in the text and we have emphasized the description of previous experiments in the text.

3. Regarding result 2, the author should summarize and improve the group differences in glucose metabolism-related indicators that appear in this section, and discuss them further, so as to make the setting of standard diets and high-fat diets more meaningful;

We agree with this reviewer that the appearance of the plots was not optimal and the labeling of certain experimental groups was relatively similar and confusing. We have changed the color code of all plots and we have described better in the figures the experiments performed to facilitate understanding. We have also summarized the description of the results of metabolic tests shown in figure 2 and we have discussed in depth the meaning of differences observed in the standard healthy diet and the obesogenic high fat diet. In this sense, we emphasized that improvements in metabolic health were occurring in HFD-fed mice treated with SB-204990, while the effects of SB-204990 in STD-fed mice were detrimental for glucoregulation. These results are similar to results of mice treated with the mTOR inhibitor rapamycin. In this line, mice treated with SB-204990 fed with healthy diet exhibited restricted levels of 4E-bp1 phosphorylation, a well-known marker of mTOR activation, suggesting that reduced mTOR signaling could contribute to the metabolic phenotype of mice treated with SB-204990 when fed with healthy STD. We have included this rationale in the description of results and discussion.

4. In terms of compound selectivity, can the same findings be reproduced in FDA-approved ACLY inhibitor?

We have performed additional experiments using bempedoic acid, the FDA-approved ACLY inhibitor. Similar to SB-204990, bempedoic acid increases hepatocyte lipid content, reduces MTT metabolic activity

and reduces the maximal oxygen consumption rate and the extracellular acidification of the media. Therefore, our results using bempedoic acid and SB-204990 suggest that Acly inhibition produces these effects at cellular level. These data have been included and discussed in the manuscript (Figure 7E-F and Figure S7D-E).

Minor comments:

1. The reference format is confusing, and it is recommended to refer to the journal for revision;

We have incorporated all requirements described in the submission guidelines of Communications Biology. According to the submission guidelines Communications Biology uses standard Nature referencing style. We have updated the references using Nature referencing style.

2. The language description is obscure. It is recommended to modify and polish ;

We apologize for the style we have used for the explanation of certain experiments. We did an effort aiming to explain complex experiments shown in our work using the maximum degree of detail. We now went over the whole manuscript in order to detect descriptions that were difficult to follow/understand and have modified and polished the language to make it clearer and more amenable.

Reviewer #2 (Remarks to the Author):

In this study, Alejandro Sola-García et al showed the effects of Acly inhibition on aging. They first demonstrated that the restriction of Acly could refrain intrinsic aging processes, as well as in flies and yeast, further indicating a conserved roles. In molecular level, they showed an independent role of Acly inhibition on histone acetylation, and some central metabolic pathway, as well as AMPK signaling were altered upon SB-204990 treatment. They also showed Acly inhibition negatively affects the mitochondrial function, as indicated with altered OCR and ECAR. I have some recommendations might help the authors to improve their work.

1. I don't think Acly inhibition makes an effective role in physiological aging, as the authors also use HFD mice as the experimental materials. So I suggest the authors should also mention high glucose metabolism in the title and they are suggested not to overclaim their conclusion.

We would like to thank this reviewer for his/her constructive comments. We agree with this reviewer since our work included a high fat diet treatment in mice. We believe that specific effects in glucoregulation must be stated in the title of the manuscript. Therefore, we propose the new title “Metabolic reprogramming by Acly inhibition using SB-204990 alters glucoregulation and modulates molecular mechanisms of aging”. Moreover, in order to convey a clear message, we have toned down our conclusions and we have extended the description of the limitations of the study. Additionally, we have better dissected the effects on mice fed with a high fat diet versus effects on healthy fed mice. In this line, we have emphasized that these effects must be taken into consideration in any potential further study using SB-204990.

2. For the independent role of histone acetylation, the authors should explain in the discussion, since it is quite different as previously reported.

We agree with this reviewer that previous research, including seminal reports of Dr. Wellen's laboratory, on experimental models lacking Acly (mostly at cellular level) have shown that, upon Acly deficiency, histone acetylation is reduced¹. However, several reports from the same authors have shown, in cell cultures and mouse tissues, that reduced expression of Acly does not always produce a significant reduction on histone acetylation^{2,3}. In these reports (Figure 7H-J of reference² and Figure Supplemental 2F for reference³), the lack of Acly expression in adipose tissue did not alter histone acetylation levels in the majority of lysine residues in several fat depots, as well as in liver tissue², while histone acetylation levels were also unaffected in Acly deficient macrophages stimulated or not with lipopolysaccharide³. We agree with the reviewer that a discussion that includes hypothesis leading to the lack of differences in histone acetylation in mice and cell cultures treated with SB-204990 should be included. In this line, it is tempting to speculate that alternative mechanisms independent of Acly compensate for the inhibition of Acly activity promoted by SB-204990, leading to normal Ac-CoA levels and similar levels of histone acetylation. It is likely that modulations on the activity of Acs or other enzymes that can produce Ac-CoA, such as the malonyl-CoA decarboxylase, could participate in this process. We have included this rationale in the discussion of the manuscript.

3. It will be good to verify the main molecular pathways identified by the authors (AMPK signaling pathway), maybe with some experiments in cell models.

We agree with this reviewer that additional data showing alterations promoted by SB-204990 in cell cultures of primary hepatocytes would add value to our study. We have performed further experiments aiming to determine effects in pathways altered in the livers of mice treated with SB-204990 for 15 weeks. Of note, restricted AMPK signaling was already shown in cell cultures of primary hepatocytes treated with 10 μ M SB-204990 for 16 hours (Figure S7D in the previous version, currently located at Figure 7L and Figure S7I). We have expanded the work in primary hepatocytes, using the same experimental approach, adding additional members of Ampk, mTOR and Sirt1 pathways. As opposed to liver tissues, we have observed that, in primary hepatocytes cultures treated with 10 μ M SB-204990 for 16 hours, alterations on markers of mTOR signaling and Sirt1 expression were absent (Figure 7L and Figure S7I). In these new experiments we have determined that SB-204990 decreases the phosphorylation on Ser 79 of the acetyl coenzyme A carboxylase, a downstream target of Ampk, which corroborates that SB-204990 reduces Ampk signaling and that restriction on Ampk signaling occurs early upon exposure to the Acly inhibitor SB-204990. Moreover, we have performed additional experiments using bempedoic acid, an FDA-approved Acly inhibitor, in cultures of primary hepatocytes. The experiments indicate that bempedoic acid mainly

recapitulates the effects of SB-204990 in primary hepatocytes (increases in hepatocyte lipid content, reductions MTT metabolic activity and reduction in the maximal oxygen consumption rate and the extracellular acidification of the media) (Figure 7E-F and Figure S7D-E). These results further substantiate that Acly inhibition produce these effects at cellular level. Lastly, we have also performed experiments using siRNA Acly following the recommendations of reviewer 3 (see specific response to reviewer 3 for a detailed description of the data). The experiments showed that siRNA Acly inhibition was effective (Figure 3A in this document) and that lipid accumulation was increased in siRNA Acly primary hepatocytes, mimicking the effects of SB-204990 (Figure 3B in this document). Remarkably, siRNA Acly-transfected primary hepatocytes treated or not with SB-204990 exhibited high lipid content when compared to untreated siRNA control hepatocytes (Figure 3B in this document). However, we have to state that the appearance of the cells was not optimal when cells were processed (Figure 3C in this document). This suggests that either the length of treatment (40 hours post-isolation when cells were processed) or chemicals used to interfere gene expression detrimentally affect primary hepatocytes. Therefore, we believe that these experiments should not be included in the manuscript.

4. The authors should improve the quality of the figures and adjust the color if possible.

According to the suggestion of the reviewer, we have changed all figures in the manuscript to include dot plots with a more distinctive color code.

Reviewer #3 (Remarks to the Author):

In the submitted manuscript, the authors investigate the downstream consequence of ATP citrate lyase (ACLY) inhibition. To this end, they employ SB-204990, a widely used cell permeable gamma-lactone prodrug with a well-defined pharmacokinetics and biological modulatory activity on lipid and cholesterol biosynthesis. The authors initially explore the impact of SB-204990 on chronological aging of *Saccharomyces cerevisiae* as well as on *Drosophila melanogaster* survival and locomotor activity. Then, they report the metabolic consequence of SB-204990 treatment on wild type C57BL/6 males. By performing metabolomics, transcriptomics and proteomics, the authors delineate the metabolic changes associated with SB-204990 treatment. The authors claim that their findings reveal novel molecular pathways linked to aging, which may be relevant to better understand metabolic diseases.

Overall, the manuscript requires a substantial revision. The submitted data are not well connected and sometime their biological meanings suffer of misinterpretation. While the effect of SB-204990 on mouse metabolism could be of interest, other aspects are inconclusive and in part redundant with previously published findings. Moreover, the relevance of these data on aging and health span is poorly supported. In my opinion, the manuscript requires a substantial work to be considered as a good candidate for Communications Biology.

Major points of concerns and comments:

1. In Fig.1, the authors describe glucose and insulin levels in young vs old mice. Moreover, they report expression changes in ACLY in mouse tissues. From these datasets, they test the impact of ACLY inhibition on longevity pathways using yeast and flies. As a reader, the two parts are completely unrelated. To better connect these multiple datasets, the authors should test the effect of SB-204990 on body weight, glucose tolerance and insulin levels in young vs old mice. This would give a rationale to run some experiments in yeast and *Drosophila melanogaster*.

*We would like to thank this reviewer for his/her valuable comments. These are excellent suggestions that should be pursued in the future. In this report we first described alterations on glucose metabolism that occur in old vs. adult and old vs. young mice. We determined that certain alterations in glucose metabolism are associated with increased hepatic Acly expression/enzymatic activity. Moreover, independent research performed in *Drosophila* has shown that heterozygous Acly mutant flies exhibit longer lifespan when compared to the wild type *Drosophila*⁴. Based on these data, we decided to evaluate whether long-term pharmacological inhibition of Acly using SB-204990 (a widely used Acly inhibitor) could alter longevity in wild type yeast and flies. Then, we developed a full project in mice using in vivo and ex vivo approaches to gain insights into the effects of long-term treatment using SB-204990. These observations included in the manuscript led to this research. In this line, similar approaches (using young, middle age and old individuals and then targeting of a specific pathway) related to the topic of our work have been published in other well-respected journals⁴. Therefore, we believe that the rationale described in the text is valid. Moreover, we believe that the research proposed using young and old mice treated or not with SB-204990 and potentially under standard healthy diet, as well as an obesogenic HFD, is not required (as well as overambitious) to justify the evaluation of yeast and fly survival upon SB-204990 treatment. Regarding the proposition of the reviewer to conduct new in vivo research using young and very old mice, we would like to mention that mice shown in figure 2 were around 180 days of age at the time of initiation of SB-204990 treatment, and that, at the time of sacrifice, mice were around 300 days old. This represents a substantial longer age when compared to the majority of the in vivo research conducted in mice. Unfortunately, we do not have very old wild type mice (700 days old; 22 month-old) in our laboratory at this moment, and we could not perform the proposed research. Based on the comments of another reviewer, we have performed additional experiments in young mice (5 week-old) treated or not with SB-204990 (5-week treatment). In these experiments we have evaluated the spontaneous activity (activity and rearing), as well as metabolic activity (energy expenditure, respiratory quotient, water intake and energy intake) using metabolic cages (indirect calorimetry system). Our results indicate that young mice treated with SB-204990 for 5 weeks do not exhibit significant alterations in the parameters evaluated. These results support that metabolic effects of SB-204990 are produced later in life. In order to improve the connection of the experiments performed*

we have better explained the rationale of running experiments in yeast, flies and mice using the Acly inhibitor SB-204990.

2. The authors test SB-204990 in *Saccharomyces cerevisiae* strain BY4741. To the best of my knowledge, ATP-citrate lyase is not expressed in non-oleaginous yeast (Boulton C.A. et al, 1981; Rodriguez S. et al, 2016; Baroni M.D. et al, 2020). If so, SB-204990 has an off-target effect in Fig. 1O. Or did the author use an ACL/ACLY expressing yeast strain? Please, clarify this issue.

In our work, we used wild type BY4741 yeast, which are naturally lacking the ATP-citrate lyase. As highlighted by the reviewer, other inhibitors of the ACLY, such as hydroxycitrate share this interesting phenotype (extended survival when compared to untreated controls)⁶. It is therefore likely that other citrate metabolism pathways are affected. Remarkably, increased survival in the absence of ACLY indicate that hampering citrate metabolism using SB-204990 (as well as hydroxycitrate) confers survival advantages.

Moreover, these data support the notion that pathways catabolizing citrate are relevant to understand SB-204990 effects in yeast and potentially in flies and mammals. We believe that it is important to stress this point in the manuscript and this rationale has been included in the description of the results, in the discussion of the manuscript, and in the limitations of the study.

3. In Fig. 1P-Q, the authors employ *Drosophila melanogaster* and explore SB-204990 effects on survival and locomotor activity. Based on their submitted data, SB-204990 extends the median lifespan of a few days. Since the authors assessed only 30 animals (n=1?), a more robust statistic would require additional biological replicates. Moreover, the authors should include ACLY mutant flies and treat them with saline (or DMSO?) and SB-204990 to rule out any off-targets. As an additional remark on Figure 1, it seems that SB-204990 may influence locomotor activity only in young

Figure 1. Effective Acly gene disruption in hemizygosis induces developmental lethality in flies. Two different fly lines harboring Acly deficiency, DG23402/CyTb and WiDf(2R)BSC308/CYO were obtained from the laboratory of Dr. Cenci and were crossed to generate effective Acly gene disruption in hemizygosis according to a previous report⁵. Of note, DG23402/CyTb flies were described as ATPCLDG23042 and WiDf(2R)BSC308/CYO flies were described as Df(2R) in Dr. Cenci's manuscript. (A) Expected mendelian ratios for the cross of DG23402/CyTb and WiDf(2R)BSC308/CYO for female and male flies. (B-C) Results obtained from these crosses. Of note the genotype CyTb/CYO is not expected to be viable, leading to 0% expected mendelian ratio. The genotype DG23402/WiDf(2R)BSC308 would produce the phenotype red eyes and straight wings. The genotype DG23402/CYO would produce the phenotype curly wings red eyes. The genotype CyTb/WiDf(2R)BSC308 would produce curly wings white eyes. (B) Females. n = 66. (C) Males. n = 78. Results indicate the developmental lethality of Acly mutant hemizygous flies (DG23402/WiDf(2R)BSC308).

animals. If so, SB-204990 has little impact on age-related processes, which would undermine some of the authors' conclusions.

As stated in the experimental methods, ~30 flies per experimental group were included in the longevity assays in flies. This “n” is limited, but it was still sufficient to determine significant differences in life expectancy. In mice, the complete deletion of Acly results in lethality⁷. In flies, effective Acly gene disruption in homozygosis or hemizygosis also results in lethality or very compromised viability⁵. These data highlights the importance of Acly in physiology. Therefore, given the profound phenotype of effective Acly gene disruption in flies, off target effects of SB-204990 could not be properly evaluated in these flies. Another previous report has already determined that reduced Acly expression/activity (heterozygous Acly mutant flies showing 20% reduction on enzymatic Acly activity when compared to wild type Acly activity) extends lifespan in flies⁴. These data support our findings using SB-204990 in fly longevity and suggest that a mild inhibition of Acly activity would lead to longer lifespan in flies. We consider that off target effects of SB-204990 could not be evaluated using heterozygous Acly flies since these flies only have ~20% reduction in Acly enzymatic activity. In order to provide further evidences indicating the critical role of Acly and restricted Acly activity in flies, we have attempted to perform the proposed research (evaluation of effects in longevity in flies exhibiting effective Acly gene disruption in hemizygosis) treating these flies or not with SB-204990. We have obtained the Acly mutant flies from the aforementioned authors⁵. We have confirmed that flies harboring effective Acly gene disruption in hemizygosis are not viable (see Figure 1 in this document), which has impeded us to determine the likely existence of off-target effects of SB-204990.

As suggested by this reviewer, it is possible that off-target effects of SB-204990 could occur in any experimental model (yeast, flies and mice) or even in humans. In this regard, several potential geroprotective agents have been investigated, and the main targets have been proposed. For example, metformin, resveratrol and rapamycin are known to have effects independent of mitochondrial complex I, mTOR and Sirt1, respectively. In order to avoid any misinterpreted conclusion, we have emphasized in the limitations of the study that we described and we have interpreted the physiological effects of the widely-used Acly inhibitor SB-204990, which could have off-target or indirect cellular and physiological effects. We agree with the reviewer that the lack of changes in spontaneous mobility of flies treated with SB-204990 late in the longevity assay might reflect that these flies have a similar health status at that age when compared to untreated control flies. In order to be precise, we have avoided statements regarding the effects of SB-204990 in healthspan in the text.

4. I am puzzled by Fig. 3. It seems clear that SB-204990 has a negative impact on lipid metabolism in animals kept under standard diet. Thus, SB-204990 would not be a recommended treatment to maintain

healthy lifespan in the majority of non-obese people, thereby undermining all the line of arguments supporting the use of ACLY inhibitors to promote longevity.

We agree with this reviewer that mice fed with healthy diet treated with SB-204990 exhibited markers of impaired metabolic control (mainly on insulin resistance, and glucoregulation). Remarkably, this phenotype is similar to the effects of mice treated with rapamycin, one of the most widely chemical used in aging research. It is actually possible that certain effects of SB-204990 might be mediated by restricted activity of the mTOR, since we observed restricted mTOR signaling (e.g. reduced phosphorylation levels of pThr 37/46 4E-bp1). We believe that SB-204990 might extend lifespan, while producing mild impairments in metabolic control in individuals fed with healthy diet. Based on our data, it is also likely that SB-204990 might not affect other aspects of health, such as neurocognitive function or motor function in healthy-fed mice. Despite the merits of SB-204990 treatment in HFD-fed mice, we agree with this reviewer that it is important to clearly state the detrimental effects on glucoregulation of SB-204990 and potentially other Acly inhibitors in healthy fed individuals. We have changed the title of the manuscript stating the alterations in glucoregulation, and we have avoided statements regarding the effects of SB-204990 in healthspan in the text. Moreover, we have emphasized that further research is required prior any potential translation of our work in any physiological condition (e.g. obese or non-obese individuals).

5. The authors claim that SB-204990 may influence physical performance in mice under HFD, which is expected if mice weight less. Surprisingly, HFD does not cause spatial memory deficits (Fig. S3), which is against what largely reported in literature. It would be recommended to review the experimental protocols and then draw the conclusions. In addition, the authors should include some histological analysis of brains from HFD mice, treated or untreated with SB-204990. Here, the authors should determine, whether SB-204990 can reduce neuroinflammation due to HFD.

We agree with the reviewer that body weight is relevant in physical performance test in mice. We considered important to determine and describe the effects in physical health in mice treated or not with SB-204990 using gold standard protocols such as rotarod and wire hang performance.

In our study, we performed the Barnes Maze test to determine any potential modulation on spatial memory. We agree with the reviewer that several previous studies found spatial memory deficits in animals fed with a HFD⁸⁻¹⁰. We believe that we have to consider the disparity between studies when spatial memory is evaluated. This includes, among other factors, differences in the age of the mice, length of HFD treatment, the composition of the diet (for instance, % of calories from fat). Remarkably, several studies did not find differences in spatial memory in animals fed with HFD¹¹⁻¹⁴. Of particular relevance is the study of Kesby et al.¹⁴. In this study authors used comparable conditions to our study (C57BL/6 mice, Barnes Maze test and a 4-5 months length of HFD treatment). In this paper authors specifically analyzed the effect of HFD

on spatial memory using the Barnes maze test. These authors found no differences between HFD and STD groups in latency during the training and memory retention. Authors only found some differences in the number of errors on day 4 of training, both groups demonstrated a similar capacity for spatial learning and memory storage. As stated by the authors, the conclusion of their study was that “HFD exposure did not significantly contribute to any age-related impairments in spatial cognition in either adult or aged mice”.

Following the recommendations of the reviewer we have reviewed our protocols and data on HFD dieting and Barnes Maze test. We have followed a previously established protocol for Barnes Maze test that has been highly used (more than 140 citations in Scopus)¹⁵. This protocol has already been used in our previous research¹⁶. Moreover, our laboratory has extensive experience using HFD¹⁶⁻¹⁹. In this regard, mice weight (Fig. 2B), as well as several metabolic (Fig. 2 C-L, Fig. 3 A-F) and histopathological parameters (Fig. 3 J-K) indicate that the HFD was effective. We have further analyzed our Barnes Maze experiment. Results showed in all experimental groups a significant decrease in the escape latency between the beginning of the training (day 2) and test days 5 and 12, demonstrating that learning in the Barnes Maze test was effective using this specific protocol (see Figure 2 in this document). However, no differences were found between STD vs. HFD, STD vs. STD-SB, and HFD vs. HFD-SB during training and memory tests. These results indicate that neither HFD nor SB-204990 have effects on this specific test under these experimental conditions. It is likely that 12-13 weeks on the HFD provided were not sufficient to produce a spatial memory deficit when mice started the HFD treatment at 26 weeks of age (quite late in life, when compared to the majority of published studies). As suggested by the reviewer, we were very interested to determine potential modulations on neuroinflammation in the mice used in our study. Of note, in order to conduct these experiments, we opted to run western blots on neuroinflammatory markers since tissues were not perfused at the time of sacrifice and tissues were flash frozen in liquid nitrogen (freezing damage would be expected, impeding immunohistochemistry analyses). Our results indicate brain lysates of untreated HFD-fed mice exhibit a significant increase in protein levels of the cytokine Ccl2 when compared to STD group (Figure S3G). Remarkably, brain lysates of HFD-SB mice showed similar levels of Ccl2 protein expression when compared with the untreated STD-fed group and significant reduced expression when brain lysates of HFD-SB mice were compared to the HFD untreated control. These results indicate that, despite the lack of effects on neurocognitive tests *in vivo*, certain pro-inflammatory processes were already promoted by HFD in untreated mice. We also evaluated expression levels of Gfap (astrocytes), Caspase 3 (pro-apoptotic) or other pro-inflammatory markers such as Tnfa and Nfkb (Figure S3G). The evaluation of these markers did not show significant difference in any experimental group, indicating that alterations in these markers were undetectable at the time of sacrifice (41 weeks of age). We have included the western blots and densitometry quantifications in Figure S3G and we have included these experiments in the manuscript.

6. The authors claim that ACLY inhibition is not associated with global modulation of histone acetylation. While the western blots may be conclusive and supporting the authors' arguments, it is not possible to rule out that transcriptional changes are due to histone posttranslational acetylation at target genes.

We agree with this reviewer that modulations in histone acetylation in target genes might occur. This rationale has been emphasized including the following statement "it is not possible to rule out that transcriptional changes are due to changes in histone acetylation in the promoter of target genes coding for master regulators of cellular metabolism. These potential changes could orchestrate metabolic arrangements that could contribute to the physiological effects of SB-204990."

7. Seahorse experiments: were the traces normalized to the number of cells or ug protein (as in Fig. 7G)? The authors wrote that "maximal OCR was diminished in cells cultured on high glucose (25 mM) treated with SB-204990". Fig. 7E shows a decreased basal respiration and maximal response in LG-SB (empty circle) compared to LG (full circle). Based on figure legend S2, it is reported that Lg: low glucose, Hg: high glucose. Could the authors clarify this discrepancy? Regarding the ECAR, I would strongly advice the authors to review their data. It may be helpful to use the glycolysis stress test to have a better picture of glycolysis in their cultured cells.

We thank this reviewer for his/her thoughtful revision. In the experiments performed using primary hepatocytes results were normalized to the number of cells (5000 cells per well). Seahorse experiments using isolated mitochondria were normalized to protein content. This information has been included in figure legends.

We apologize that the description of the data was unclear.

The results of this experiment show significant decreases in basal and FCCP-induced maximal respiration in the LG-SB vs. LG comparison (standard culture conditions; 5 mM glucose). The same plot also shows decreased FCCP-induced maximal OCR in the HG-SB vs. HG comparison (high glucose; 25 mM glucose). In order to facilitate understanding we have better described the results obtained and we have added pertinent information of the experimental conditions in the legend of the figure. We have realized that we had included two different definitions of low-glucose and high-glucose in Figure S2 and Figure 7. In order to avoid misunderstandings, we have eliminated the definition of low glucose and high glucose in the experiments using pancreatic islets (Figure S2). For experiments shown on Figure 7 low glucose represents 5 mM glucose (standard culture conditions) and high glucose represents 25 mM glucose. We have re-written figure legends stating the actual glucose concentration in the experiments. Then, the definition of low or high glucose is specifically used for experiments shown in Figure 7 and Figure S7 (experiments performed in primary hepatocytes).

We have reviewed the data on the ECAR. The data shown in the figures is correct. Our results in Seahorse experiments using primary hepatocytes are very similar to available data in the literature²⁰. Our data reflects that SB-204990 produced reductions in ECAR when cells were exposed to Oligomycin or FCCP (treatments that compromise mitochondrial ATP production). We hypothesized that reduced ECAR is caused by restricted glycolysis in cells treated with SB-204990. We agree with the reviewer that performing additional experiments using the glycolysis stress test would add scientific meaning to our work. We have performed the experiments proposed. Data generated indicate that primary hepatocytes treated with SB-204990 under standard conditions (5 mM glucose) exhibited a marked reduction in the ECAR in the glycolysis stress test (Figure 7I). A similar effect was observed in primary hepatocytes treated with SB-204990 and cultured in high glucose (25 mM) when compared to primary hepatocytes cultured in high glucose (25 mM) or standard glucose concentration (5 mM). These results further substantiate that SB-204990 inhibits glycolysis. These experiments have been included in Figure 7I and data has been described and discussed.

8. In Fig. 7, the authors report a large number of immunoblots for various kinases and their downstream targets. While I appreciate the authors' effort to dissect ACLY-dependent signaling pathway, it is very difficult to follow their line of arguments. The data are confusing and it is not possible to understand how ACLY influences AMPK and mTOR signaling. It would be important to re-organize these large datasets in a more understandable manner. Moreover, I would recommend to employ genetic approaches (e.g., shRNA) to mechanistically support their conclusions.

We agree with the reviewer that a different approach on the description of this part of the manuscript could facilitate understanding. Following the advice of the reviewer we have modified this section. It is difficult

to define how SB-204990 influences altered pathways using mouse tissues, and we have opted to first describe expression levels of kinases and downstream targets in liver tissue. We have dissected the explanation of these data starting with the description of the western blots performed in liver tissue in STD and HFD treated or not with SB-204990 for 15 weeks. We describe that there is a reduction on AMPK signaling in mice treated with SB-204990 fed with STD and HFD (Figure 7K and Figure S7F). We also describe that markers of restricted mTOR signaling and increased Sirt1 expression are found in STD-SB-204990 mice, while unaltered mTOR signaling and increased Nmnat expression are found in HFD-SB-204990 mice. Lastly, we also added the evaluation of expression levels of markers of mitochondrial content (reduced in STD-SB mice), mitochondrial β -oxidation (increased STD-SB and HFD-SB groups) and oxidative/cellular damage (reduced in STD-SB and HFD-SB groups) (Figure 7K and Figure S7F-H). In order to advance the understanding of the effects of the Acly inhibitor SB-204990 we subsequently have described effects occurring early upon treatment with SB-204990 in cell cultures of primary hepatocytes (Figure 7L and Figure S7I). We determined that SB-204990-induced increases on lipid accumulation and decreases on metabolic activity (MTT test) do not occur when cells are treated simultaneously with Ampk activators (metformin and AICAR) (Figure 7M-O and Figure S7O). In addition, new experiments show that restricted Ampk and Acc phosphorylation occur in these hepatocytes (Figure 7L and Figure S7I). We then describe separated experiments using SB-204990 and activators/inhibitors of mTOR and folate cycle (Figure S7O). These experiments show that alterations on lipid content or metabolic activity promoted by SB-204990 are still present when hepatocytes are treated mTOR/folate cycle activators/inhibitors. Moreover, we added several western blots indicating that expression levels of markers of mTOR signaling and Sirt1 expression are not altered in hepatocytes treated with SB-204990 for 16 hours (Figure 7L and Figure S7I). These experiments indicate that early responses to SB-204990 rely preferentially on the modulation of Ampk activity. We have included these experiment, descriptions, and rationale in the manuscript.

We agree with the reviewer that, to further support the data in this figure, additional research using cell cultures would help us to understand the effects of the Acly inhibitor SB-204990. However, experiments using genetic approaches are challenging in primary cultures (the cell cultures used in this figure) since the viability of primary cultures is very time-limited and genetic approaches have to be implemented for a relatively long time to be effective. Moreover, chemicals used to interfere gene expression might be especially toxic for primary cells. We performed experiments using siRNA interference based on our previous experience using this technology and the duration of primary hepatocyte cell cultures. In particular, we have performed experiments using primary hepatocytes cultures of 40 hours of duration. Primary hepatocytes were siRNA interfered 4-hours post-isolation. We initiated SB-204990 treatment at 24 hours post-isolation and cells were processed at 40 hours post-isolation (the duration of SB-204990

treatment was set at 16 hours, in agreement to the rest of the experiments performed in our study). The experiments showed that siRNA Acly inhibition was effective (Figure 3A in this document) and that lipid accumulation was increased in siRNA Acly primary hepatocytes, mimicking the effects of SB-204990 (Figure 3B in this document). Remarkably, siRNA Acly interfered primary hepatocytes treated or not with SB-204990 exhibited high lipid content when compared to untreated siRNA control hepatocytes (Figure 3B in this document). However, we have to state that the appearance of the cells was not optimal when cells were processed (Figure 3C in this document). This suggests that either the length of treatment (40 hours post-isolation when cells were processed) or chemicals used to interfere gene expression detrimentally affect primary hepatocytes. Therefore, we believe that these experiments should not be included in the manuscript. In addition, following the advice of another reviewer, we have also treated primary hepatocytes with bempedoic acid, an FDA-approved Acly inhibitor used for the treatment of hypercholesterolemia. Our results indicate that bempedoic acid has similar effects to SB-204990 (MTT, Oil Red O and Seahorse experiments), indicating that different Acly inhibitors produce similar effects in primary hepatocytes (Figure 7E-F and Figure S7D-E). These experiments support our main observations.

Figure 3. Experiments in primary hepatocytes using siRNA Acly. (A-C) Primary hepatocytes were siRNA interfered for Acly expression for 40 hours and were treated for 16 hours (24 hours after siRNA interference) with 10 μM SB-204990. (A) Acly expression levels determined by real time PCR. $n = 5$ per group. Two-way ANOVA. (B) Quantification of Oil red O staining of primary hepatocytes treated with 10 μM SB-204990. $n = 13$ per group. Two-way ANOVA. (C) Representative images of siRNA interfered primary hepatocytes stained with oil red O treated with 10 μM SB-204990. Scale bar 50 μm. $n = 13$ per group. Scale bar: 50μM. SB: SB-204990. NS: not significant. r.u.: relative units. Data shown are the means \pm SEM. * $p < 0.05$ on any experimental group vs. siRNA control Ut.

9. The authors claim that metformin and AICAR partially rescue lipid content promoted by SB-204990. Do they rescue also the mitochondrial defects?

It is known that metformin and AICAR target mitochondrial metabolism, reducing the OCR^{21,22}. Therefore, we anticipated that both, metformin and AICAR, might not increase mitochondrial OCR in cells treated or not with SB-204990. In order to provide the data suggested by the reviewer, we have performed additional experiments using AICAR and metformin in primary hepatocytes treated or not with SB-204990 (Figure S7K-N). As expected, metformin and AICAR did not increase the OCR in any experimental condition. Actually, metformin and AICAR further exaggerated mitochondrial effects in cells treated with SB-204990, indicating that, at the doses tested, the effects of SB-204990 and Ampk activators are synergistic. These data are now presented in Figure S7K-N and discussed.

References:

- 1 Wellen, K. E. *et al.* ATP-citrate lyase links cellular metabolism to histone acetylation. *Science* **324**, 1076-1080, doi:10.1126/science.1164097 (2009).
- 2 Zhao, S. *et al.* ATP-Citrate Lyase Controls a Glucose-to-Acetate Metabolic Switch. *Cell reports* **17**, 1037-1052, doi:10.1016/j.celrep.2016.09.069 (2016).
- 3 Baardman, J. *et al.* Macrophage ATP citrate lyase deficiency stabilizes atherosclerotic plaques. *Nature communications* **11**, 6296, doi:10.1038/s41467-020-20141-z (2020).
- 4 Peleg, S. *et al.* Life span extension by targeting a link between metabolism and histone acetylation in *Drosophila*. *EMBO reports* **17**, 455-469, doi:10.15252/embr.201541132 (2016).
- 5 Morciano, P. *et al.* Depletion of ATP-Citrate Lyase (ATPCL) Affects Chromosome Integrity Without Altering Histone Acetylation in *Drosophila* Mitotic Cells. *Frontiers in physiology* **10**, 383, doi:10.3389/fphys.2019.00383 (2019).
- 6 Baroni, M. D. *et al.* In *S. cerevisiae* hydroxycitric acid antagonizes chronological aging and apoptosis regardless of citrate lyase. *Apoptosis : an international journal on programmed cell death* **25**, 686-696, doi:10.1007/s10495-020-01625-1 (2020).
- 7 Beigneux, A. P. *et al.* ATP-citrate lyase deficiency in the mouse. *The Journal of biological chemistry* **279**, 9557-9564, doi:10.1074/jbc.M310512200 (2004).
- 8 Jeon, B. T. *et al.* Resveratrol attenuates obesity-associated peripheral and central inflammation and improves memory deficit in mice fed a high-fat diet. *Diabetes* **61**, 1444-1454, doi:10.2337/db11-1498 (2012).
- 9 Vinuesa, A. *et al.* Juvenile exposure to a high fat diet promotes behavioral and limbic alterations in the absence of obesity. *Psychoneuroendocrinology* **72**, 22-33, doi:10.1016/j.psyneuen.2016.06.004 (2016).
- 10 Spencer, S. J. *et al.* High-fat diet and aging interact to produce neuroinflammation and impair hippocampal- and amygdalar-dependent memory. *Neurobiology of aging* **58**, 88-101, doi:10.1016/j.neurobiolaging.2017.06.014 (2017).
- 11 McNeilly, A. D., Williamson, R., Sutherland, C., Balfour, D. J. & Stewart, C. A. High fat feeding promotes simultaneous decline in insulin sensitivity and cognitive performance in a delayed matching and non-matching to position task. *Behavioural brain research* **217**, 134-141, doi:10.1016/j.bbr.2010.10.017 (2011).
- 12 Pancani, T. *et al.* Effect of high-fat diet on metabolic indices, cognition, and neuronal physiology in aging F344 rats. *Neurobiology of aging* **34**, 1977-1987, doi:10.1016/j.neurobiolaging.2013.02.019 (2013).
- 13 Bah, T. M. *et al.* GPR39 Deficiency Impairs Memory and Alters Oxylipins and Inflammatory Cytokines Without Affecting Cerebral Blood Flow in a High-Fat Diet Mouse Model of Cognitive Impairment. *Frontiers in cellular neuroscience* **16**, 893030, doi:10.3389/fncel.2022.893030 (2022).

- 14 Kesby, J. P. *et al.* Spatial Cognition in Adult and Aged Mice Exposed to High-Fat Diet. *PLoS one* **10**, e0140034, doi:10.1371/journal.pone.0140034 (2015).
- 15 Patil, S. S., Sunyer, B., Hoger, H. & Lubec, G. Evaluation of spatial memory of C57BL/6J and CD1 mice in the Barnes maze, the Multiple T-maze and in the Morris water maze. *Behavioural brain research* **198**, 58-68, doi:10.1016/j.bbr.2008.10.029 (2009).
- 16 Lopez-Noriega, L. *et al.* Inadequate control of thyroid hormones sensitizes to hepatocarcinogenesis and unhealthy aging. *Aging* **11**, 7746-7779, doi:10.18632/aging.102285 (2019).
- 17 Lorenzo, P. I. *et al.* The metabesity factor HMG20A potentiates astrocyte survival and reactive astrogliosis preserving neuronal integrity. *Theranostics* **11**, 6983-7004, doi:10.7150/thno.57237 (2021).
- 18 Alvarez-Amor, L. *et al.* Extra virgin olive oil improved body weight and insulin sensitivity in high fat diet-induced obese LDLr^{-/-}.Leiden mice without attenuation of steatohepatitis. *Scientific reports* **11**, 8250, doi:10.1038/s41598-021-87761-3 (2021).
- 19 Martin-Montalvo, A. *et al.* Cytochrome b5 reductase and the control of lipid metabolism and healthspan. *NPJ aging and mechanisms of disease* **2**, 16006, doi:10.1038/npjamd.2016.6 (2016).
- 20 Torres, A., Noriega, L. G., Delgadillo-Puga, C., Tovar, A. R. & Navarro-Ocana, A. Caffeoylquinic Acid Derivatives of Purple Sweet Potato as Modulators of Mitochondrial Function in Mouse Primary Hepatocytes. *Molecules* **26**, doi:10.3390/molecules26020319 (2021).
- 21 Spangenburg, E. E., Jackson, K. C. & Schuh, R. A. AICAR inhibits oxygen consumption by intact skeletal muscle cells in culture. *Journal of physiology and biochemistry* **69**, 909-917, doi:10.1007/s13105-013-0269-0 (2013).
- 22 Martin-Montalvo, A. *et al.* Metformin improves healthspan and lifespan in mice. *Nature communications* **4**, 2192, doi:10.1038/ncomms3192 (2013).

Sincerely yours,

Alejandro Martín-Montalvo

Principal Investigator

CABIMER

Av/ Américo Vespucio 24, Seville, CP41092, Spain.

Phone.: 0034 954 46 74

Email address: alejandro.martinmontalvo@cabimer.es

Reviewers' comments:

Reviewer #1 (Remarks to the Author):

The authors have performed additional experiments which addressed previous concerns. The reviewer has several additional comments for consideration:

1. Figure 2, differential effect of SB compound on chow diet and high fat diet feed mice need justification.
2. Figure 3, levels of ALT and AST are good markers of liver injury in HFD treated mice, any effect of SB on diet induced steatohepatitis?
3. Figure 4, any effect on H3K27ac? global expression of histone markers does not necessarily say histone binding to target gene affected.
4. Figure 7, the effect of SB on hepatocyte lipid accumulation is hard to understand, as theoretically, ACLY inhibition would limit de novo lipogenesis and reduce lipid/TG accumulation.
5. ACLY inhibition always come with ACSS2 activation, any role of ACSS2 in SB compound mediated effects?

Reviewer #2 (Remarks to the Author):

The authors have revised this work appropriately, I have no further questions.

Reviewer #3 (Remarks to the Author):

In their revised manuscript, the authors adjusted some of the figures and text as recommended by the three referees. Although I appreciate their efforts in responding to some of my comments, some critical aspects have not been fully addressed with new experimental evidence. I can understand that some experiments with mice may be time consuming, however there are a couple of things that cannot be ignored and must be rectified because they give a misleading message to the readers.

Q2A. One of my comments was about SB-204990 in yeast. The authors confirmed that ATP-citrate lyase is not expressed in non-oleaginous yeast, meaning that SB-204990 modulates yeast lifespan independently of ATP-citrate lyase. If it is through citrate metabolism as the authors suggested in the response, it must be proven experimentally. Based on these arguments, the authors cannot write in the Abstract that "pharmacological inhibition of Acl_y using SB-204990 is effective prolonging lifespan in yeast". It is wrong and misleading.

Q2B. I asked the authors to refine their experiments in *Drosophila*. One issue is about the specificity of SB-204990 on ACLY in flies. The authors made a series of arguments about homozygous vs heterozygous. However, they mentioned a paper (Peng Shahaf et al, 2016) in which lifespan assays were performed with +/atpcl mutant flies (Figure 4B). I asked and ask again to treat these mutant flies (or flies in which atpcl is downregulated) with SB-204990 to determine if the effect is dependent on ACLY. Otherwise, their statement in the Abstract ("pharmacological inhibition of Acl_y using SB-204990 is effective prolonging lifespan in flies)" is incorrect.

Q2C. The authors employed <30 flies for their lifespan assay. The experiment was repeated only once, unless I misunderstood their Figure legends. Since *Communication Biology* follows the Good Scientific Practice, I expect that the authors include additional biological replicates (n>3).

Q7. I expressed my concern about the ECAR traces. Respectfully, measurements in figure 7I are puzzling for the following reasons: 1) Cells do not respond to additional glucose; 2) Upon oligomycin treatment, only HG-SB cells show a slight increase in ECAR; 3) Upon exposure to 2-DG, some cells (e.g., HG) have a ECAR that is much lower than the basal levels. These are not ECAR measurements

generally accepted by colleagues working with a Agilent Seahorse. I kindly encourage the authors to revise their experimental protocol.

Q8. I asked to adjust the immunoblots in Figure 7K-L because it was difficult to follow their arguments. I still believe that readers may have difficulties to appreciate these large datasets. The authors wrote that pThr172 Ampk was reduced, however phosphorylation of the target enzyme Acc has a different trend in the different animal cohorts (Figure S7F). Does it mean that Ampk activity is differentially regulated in STD vs HFD mice? The authors suggested that mTOR activity is dysregulated in STD-SB treated mice, however that is not the case for S6, since pSer235/236 levels do not reach statistical significance.

Dear Dr. Rogers,

We are pleased to resubmit our work entitled "**Metabolic reprogramming by Acly inhibition using SB-204990 alters glucoregulation and modulates molecular mechanisms of aging**" to Communications Biology. We have addressed the comments from the three reviewers. We include a point by point response, marked in italic, for all comments provided:

Reviewer #1 (Remarks to the Author):

The authors have performed additional experiments which addressed previous concerns. The reviewer has several additional comments for consideration:

1. Figure 2, differential effect of SB compound on chow diet and high fat diet feed mice need justification.

We would like to thank the reviewer for the new revision of the manuscript. We have included an extended discussion of differentially altered pathways in mice under the different feeding conditions (STD or HFD) that could contribute to the phenotypic changes observed in our study. We have opted to focus our discussion on differential alterations in mTOR signaling in STD-fed and HFD-fed mice treated or not with SB-204990, since it is known that blockade of mTOR (pharmacological or genetic) results in compromised glucoregulation in mice fed with healthy diets, as observed in mice treated with SB-204990 fed with STD¹⁻³. We have discussed that we found reduced levels of 4E-BP1 phosphorylation, a marker of mTOR activity, in the livers of mice treated with SB-204990 fed with healthy STD. This alteration did not occur in the livers of mice treated with SB-204990 when mice were fed with HFD. Remarkably, phosphorylated levels of S6, another target of mTOR, did not reach statistical significance neither in mice fed with SB-204990 with STD nor in mice fed with SB-204990 with HFD. These results suggest that specific alterations in mTOR signaling occurring in mice fed with a healthy STD might predispose to compromised glucoregulation. Moreover, the fact that not all markers of mTOR activity show significantly altered levels in STD-SB mice suggests that specific branches of mTOR signaling are altered by SB-204990 in mice fed with STD (but are unaltered in mice fed with HFD). These effects might indicate that additional regulatory processes, such as activation of phosphatases or other kinases that regulate targets of mTOR could be differentially activated in the different feeding conditions/treatments used in our study. This rationale has been included in the discussion of manuscript.

2. Figure 3, levels of ALT and AST are good markers of liver injury in HFD treated mice, any effect of SB on diet induced steatohepatitis?

Our histological analysis clearly showed that lipid infiltration was reduced in the liver of mice treated with SB-204990 when mice were challenged with HFD. Therefore, it is tempting to speculate that SB-204990 protects from steatohepatitis. We have evaluated ALT and AST levels in the serum of the mice of the study. Our results indicate that serum ALT levels were increased in mice fed with HFD when compared to STD-fed mice, while serum AST levels were not altered in HFD-fed mice. These makers of hepatic damage remained unaltered in mice treated with SB-204990 in either STD or HFD (Figure 3L-M). Of note, ALT levels in HFD-SB mice were not significantly altered when compared to untreated STD-fed mice, suggesting an intermediate phenotype. These results indicate that certain degree of hepatic damage was promoted in untreated HFD-fed mice after 14 weeks of HFD dieting and that SB-204990 did not produce hepatotoxicity in either STD or HFD. These results have been incorporated into the manuscript.

3. Figure 4, any effect on H3K27ac? global expression of histone markers does not necessarily say histone binding to target gene affected.

In the previous submitted version of the manuscript we included acetylation levels of 5 histone lysine residues, showing the lack of differences in histone acetylation levels in the livers of mice treated with SB-204990 fed with either STD or HFD. As suggested, we have evaluated the levels of H3K27Ac in the livers of the mice used in our study. Results indicate that SB-204990 did not alter acetylation levels on this specific residue in mice treated with SB-204990 fed with in either STD or HFD. These results have been included in Figure 4B-C. We agree with the reviewer that the lack of differences in global acetylation levels in any specific lysine does not necessarily implicate differential binding of transcriptional regulatory proteins to specific target genes. Our main conclusion is that SB-204990 is not altering global levels of histone lysine acetylation in the residues tested. We agree with the reviewers that alterations in histone acetylation levels in specific regions of the genome could be occurring, and those effects could be relevant. We have emphasized this rationale in the discussion of the manuscript.

4. Figure 7, the effect of SB on hepatocyte lipid accumulation is hard to understand, as theoretically, ACLY inhibition would limit de novo lipogenesis and reduce lipid/TG accumulation.

We agree with the reviewer that, in principle, upon Acl_y inhibition, one would expect reduced levels of de novo lipogenesis, which could lead to reduced lipid accumulation. However, our observations indicate that SB-204990 and bempedoic acid increase lipid content in hepatocytes. As described in the manuscript,

similar effects (e.g. larger lipid droplets/adipocyte size) have been observed in healthy-fed mice and cell cultures depleted of Acly activity⁴⁻⁷. In this regard, we hypothesized that reduced mitochondrial function could contribute to accumulate lipids in hepatocytes treated with SB-204990 or bempedoic acid. Our experiments using primary hepatocytes described in figure 7 support this hypothesis. Moreover, one could also speculate that alternative Ac-CoA sources, such as malonyl-CoA via malonyl-CoA decarboxylase activity, could also significantly contribute to accumulate intracellular lipids in hepatocytes treated with SB-204990 or bempedoic acid. This line of arguments has been empathized in the discussion of the manuscript.

5. ACLY inhibition always come with ACSS2 activation, any role of ACSS2 in SB compound mediated effects?

We actually had evaluated the protein levels of the cytoplasmic Acetyl coenzyme A synthetase, the protein encoded by Acss2, in the livers of mice treated with SB-204990. The results indicated that protein expression levels of the cytosolic Acetyl coenzyme A synthetase are not altered neither in the livers of mice fed with STD nor in the liver of mice fed with HFD when treated with SB-204990. We have now included the evaluation of protein expression levels of Acss2 in cultures of primary hepatocytes treated with SB-204990. The results indicate that SB-204990 does not alter Acss2 expression levels. These results suggest that alterations in Acss2 expression might not play a significant role in compound-mediated effects.

This gene/protein has several alias, and we had defined this protein as the “cytoplasmic Ac-CoA synthetase (Acs, also known as AceCS1)” in the previous version of the manuscript. We have now included that the cytosolic Acetyl coenzyme A synthetase is also known as Acs, AceCS1 or Acss2. We have opted to avoid the use of the acronyms (Acs, AceCS1 or Acss2) in the text to avoid misinterpretations among the mitochondrial acetyl coenzyme A synthetase and the cytosolic acetyl coenzyme A synthetase. Expression levels of the cytosolic Acetyl coenzyme A synthetase are now defined as Acss2 in Figure 7 and Figure S7.

Reviewer #2 (Remarks to the Author):

The authors have revised this work appropriately, I have no further questions.

We would like to thank the reviewer for the evaluation of our work.

Reviewer #3 (Remarks to the Author):

In their revised manuscript, the authors adjusted some of the figures and text as recommended by the three referees. Although I appreciate their efforts in responding to some of my comments, some critical aspects

have not been fully addressed with new experimental evidence. I can understand that some experiments with mice may be time consuming, however there are a couple of things that cannot be ignored and must be rectified because they give a misleading message to the readers.

Q2A. One of my comments was about SB-204990 in yeast. The authors confirmed that ATP-citrate lyase is not expressed in non-oleaginous yeast, meaning that SB-204990 modulates yeast lifespan independently of ATP-citrate lyase. If it is through citrate metabolism as the authors suggested in the response, it must be proven experimentally. Based on these arguments, the authors cannot write in the Abstract that “pharmacological inhibition of Acl_y using SB-204990 is effective prolonging lifespan in yeast”. It is wrong and misleading.

Q2B. I asked the authors to refine their experiments in *Drosophila*. One issue is about the specificity of SB-204990 on ACLY in flies. The authors made a series of arguments about homozygous vs heterozygous. However, they mentioned a paper (Peng Shahaf et al, 2016) in which lifespan assays were performed with +/atpcl mutant flies (Figure 4B). I asked and ask again to treat these mutant flies (or flies in which atpcl is downregulated) with SB-204990 to determine if the effect is dependent on ACLY. Otherwise, their statement in the Abstract (“pharmacological inhibition of Acl_y using SB-204990 is effective prolonging lifespan in flies”) is incorrect.

Q2C. The authors employed <30 flies for their lifespan assay. The experiment was repeated only once, unless I misunderstood their Figure legends. Since Communication Biology follows the Good Scientific Practice, I expect that the authors include additional biological replicates (n>3).

We would like to thank the reviewer for the feedback. We performed experiments in yeast and flies to add value to the manuscript. However, we feel from the revision process, that these experiments are rather representing a limitation for the study, when the specific weight of these experiments in the manuscript is minimal. At this stage, we would like to propose to eliminate from the manuscript the experiments performed in yeast and flies and to adapt the description and interpretations of the manuscript to results obtained using mice (in vivo and ex vivo) and cell cultures. We have adapted the manuscript in this sense (note mainly only 2 small paragraphs, in the results and in the discussion sections, have been eliminated in the entire manuscript). Alternatively, we could also propose to adapt the interpretations of the experiments performed as the reviewer suggested (e.g. avoiding the use of the statement “pharmacological inhibition of Acl_y using SB-204990 produces in yeast or flies...”) and to avoid the use of statistical analyses (and biological interpretation) in experiments performed with a suboptimal number of replicates.

Q7. I expressed my concern about the ECAR traces. Respectfully, measurements in figure 7I are puzzling for the following reasons: 1) Cells do not respond to additional glucose; 2) Upon oligomycin treatment, only HG-SB cells show a slight increase in ECAR; 3) Upon exposure to 2-DG, some cells (e.g., HG) have a ECAR that is much lower than the basal levels. These are not ECAR measurements generally accepted by colleagues working with a Agilent Seahorse. I kindly encourage the authors to revise their experimental protocol.

As stated in the methods, we used the exact protocol described by the manufacturer (Agilent Seahorse) to run the glycolysis stress test (note that technically this assay is quite simple). We have deeply looked in the literature for experiments using the glycolysis stress test in primary hepatocytes. We only found a single report using primary hepatocytes (Figure 7B in reference⁸). The results shown in this report are similar to our observations under standard culture conditions and glucose-challenged conditions (e.g. absence of remarkable increase in ECAR in glucose-challenged primary hepatocytes of C57Bl6 mice). Remarkably, in the mentioned paper, authors did not show the effects of oligomycin and 2-deoxyglucose on primary hepatocytes, which suggest that standard patterns in ECAR in the glycolysis stress kit using other cell lines might not occur in primary hepatocytes (for instance, absence of major increases in oligomycin-challenged primary hepatocytes). We would like to emphasize that the results of the experiments proposed cannot be anticipated based on the profiles of other cell cultures. To further investigate this interesting profile, we have adapted the glycolysis stress test increasing the concentration of glucose (25 mM glucose instead of 10 mM glucose) and oligomycin (10 μ M oligomycin instead of 1 μ M oligomycin) to enhance the action of these chemicals (see Figure 1 in this document). These new results show that, in basal conditions, SB-204990 still produces reductions in ECAR when cells are cultured in standard glucose concentration (Lg; 5 mM glucose) and when cells were cultured with high glucose concentration (HG; 25 mM glucose). We observed an ~50 % increase in ECAR (still far from the 3-fold increase expected in other cell cultures described by Agilent) in all experimental conditions when glucose (25 mM) was added to the cells in the assay. Of note, differences in cells treated with SB-204990 (in LG and HG) were also significant at that time in the assay.

Figure 1. Effects of SB-204990 in glycolysis stress test in an adapted protocol that increases glucose (25 mM) and oligomycin (10 μ M) challenges. Primary cultures of murine hepatocytes were cultured with SB-204990 (10 μ M) for 16 hours and the extracellular acidification rate on glycolysis stress test was determined. $n = 4$ per group. SB: SB-204990. Glu: glucose. Oligo: Oligomycin. 2-DG: 2-deoxyglucose. Data shown are the means \pm SEM. * $p < 0.05$ LG-SB vs. LG. # $p < 0.05$ HFD-SB vs. HFD.

Interestingly, oligomycin (10 μ M) still did not enhance ECAR in any experimental condition. Actually, oligomycin reduced

ECAR in all conditions, suggesting a possible toxic effect. ECAR levels were reduced below basal ECAR levels when 2-deoxyglucose was added to the cell cultures, and differences among the experimental groups were absent at that time in the assay.

We believe that our results are indicating that primary hepatocytes have, under basal conditions, already very high levels of glycolysis. One could speculate that these cells, which are known to generate glucose via gluconeogenesis and glycogenolysis, have certain degree of glucose availability despite the fact that glucose is not added to the cell culture media. Glucose availability could facilitate the maintenance of high levels of glycolysis. Then, standard glucose (10 mM) and oligomycin (1 μ M) challenges in the glycolysis stress kit assay would not have, in primary hepatocytes, the effects observed in other cell types (increase in ECAR). The results of 2-deoxyglucose (drop on ECAR levels below basal ECAR levels) might reflect that, at that moment, glycolysis would be really inhibited by 2-deoxyglucose, leading to reduced ECAR levels. Despite these interpretations of the results generated, we believe that the fact that SB-204990 reduces glycolytic capacity has been properly supported with the experiments provided.

Q8. I asked to adjust the immunoblots in Figure 7K-L because it was difficult to follow their arguments. I still believe that readers may have difficulties to appreciate these large datasets. The authors wrote that pThr172 Ampk was reduced, however phosphorylation of the target enzyme Acc has a different trend in the different animal cohorts (Figure S7F). Does it mean that Ampk activity is differentially regulated in STD vs HFD mice? The authors suggested that mTOR activity is dysregulated in STD-SB treated mice, however that is not the case for S6, since pSer235/236 levels do not reach statistical significance.

We appreciate the effort of the reviewer to improve our work. The western blots shown determine alterations or the lack of alterations in several pathways relevant for important cellular tasks. Moreover, we did an effort to evaluate, in certain cases, several markers of the same pathway to provide a wider view of the alterations promoted by SB-204990. We have modified the description of the western blots in this section using a more simplistic but still detailed description of the data to facilitate understanding.

We agree with the reviewer that despite the fact that pThr172 Ampk phosphorylation was reduced in the liver of both, STD-SB vs. STD, and HFD-SB vs. HFD, restrictions on pSer79 Acc were specifically found in the liver of STD-SB vs. STD. It is plausible that alternative mechanisms that contribute to Acc phosphorylation could be differentially regulated in mice fed using STD or mice fed using HFD when treated with SB-204990. In this regard, we could speculate that alterations in the activity of the

phosphatases involved in the dephosphorylation of Acc, such as Pp-2a^o, could be mediating the lack of differences in Acc phosphorylation in the liver of HFD-SB- vs. HFD.

We agree with the reviewer that we only found differences in p4E-BP1 and a trend towards reduced pS6 (that did not reach statistical significance) in the liver of mice treated with SB-204990 fed with STD. We have now described in the detail that specifically phosphorylated levels of 4E-BP1 were reduced while phosphorylated levels of S6, another marker of mTOR activity, were not significantly altered. The fact that not all markers of mTOR activity show significantly altered levels in STD-SB mice suggests that specific branches of mTOR signaling are altered by SB-204990 in mice fed with STD (but are unaltered in mice fed with HFD). These effects might indicate that additional regulatory processes, such as activation of phosphatases or other kinases that regulate targets of mTOR, could be differentially activated in the different feeding conditions/treatments used in our study. We have expanded this rationale in the discussion of manuscript.

Sincerely yours,

Alejandro Martín-Montalvo

Principal Investigator

CABIMER

Av/ Américo Vespucio 24, Seville, CP41092, Spain.

Phone.: 0034 954 46 74

Email address: alejandro.martinmontalvo@cabimer.es

References:

- 1 Harrison, D. E. *et al.* Rapamycin fed late in life extends lifespan in genetically heterogeneous mice. *Nature* **460**, 392-395, doi:10.1038/nature08221 (2009).
- 2 Lamming, D. W. *et al.* Rapamycin-induced insulin resistance is mediated by mTORC2 loss and uncoupled from longevity. *Science* **335**, 1638-1643, doi:10.1126/science.1215135 (2012).
- 3 Tang, Y. *et al.* Adipose tissue mTORC2 regulates ChREBP-driven de novo lipogenesis and hepatic glucose metabolism. *Nature communications* **7**, 11365, doi:10.1038/ncomms11365 (2016).

- 4 Martínez Calejman, C. *et al.* mTORC2-AKT signaling to ATP-citrate lyase drives brown adipogenesis and de novo lipogenesis. *Nature communications* **11**, 575, doi:10.1038/s41467-020-14430-w (2020).
- 5 Migita, T. *et al.* ATP citrate lyase: activation and therapeutic implications in non-small cell lung cancer. *Cancer research* **68**, 8547-8554, doi:10.1158/0008-5472.CAN-08-1235 (2008).
- 6 Migita, T. *et al.* Inhibition of ATP citrate lyase induces triglyceride accumulation with altered fatty acid composition in cancer cells. *International journal of cancer* **135**, 37-47, doi:10.1002/ijc.28652 (2014).
- 7 Zhao, S. *et al.* ATP-Citrate Lyase Controls a Glucose-to-Acetate Metabolic Switch. *Cell reports* **17**, 1037-1052, doi:10.1016/j.celrep.2016.09.069 (2016).
- 8 Fahlbusch, P. *et al.* Adaptation of Oxidative Phosphorylation Machinery Compensates for Hepatic Lipotoxicity in Early Stages of MAFLD. *Int J Mol Sci* **23**, doi:10.3390/ijms23126873 (2022).
- 9 Gaussin, V., Hue, L., Stalmans, W. & Bollen, M. Activation of hepatic acetyl-CoA carboxylase by glutamate and Mg²⁺ is mediated by protein phosphatase-2A. *Biochem J* **316** (Pt 1), 217-224, doi:10.1042/bj3160217 (1996).

REVIEWERS' COMMENTS:

Reviewer #1 (Remarks to the Author):

The manuscript is acceptable in the current form.

Reviewer #3 (Remarks to the Author):

The authors have performed additional experiments that addressed (or not) my comments. Overall, I appreciate their efforts. I agree with the publication of their work.

Two important final considerations:

1. Some of their data are still puzzling. However, since Communications Biology uses a transparent peer-review process, my concerns as well as the authors' arguments will be available for all the readers, which should help to better interpret the results.

2. The authors decide to remove the data obtained using yeast and Drosophila. I support their decision, since some of the conclusions were based on too preliminary observations. However, by removing these findings, the authors dilute the evidence that SB-204990 influences aging processes. Thus, some of their claims are incorrect. I would recommend that the authors adjust the title (it should be written: "Metabolic reprogramming by Acly inhibition using SB-204990 alters gluco-regulation and modulates molecular mechanisms associated with aging") and the abstract (it should be written "SB-204990 plays an unprecedented role in the regulation of molecular mechanisms associated with aging") of their manuscript.